# Epigenomic landscape of single vascular cells reflects developmental origin and disease risk loci

Chad S Weldy [1,2✉], Soumya Kundu[3,4], João Monteiro [1], Wenduo Gu [1], Albert J Pedroza [5], Alex R Dalal [5], Matthew D Worssam [1], Daniel Li [1], Brian Palmisano [1], Quanyi Zhao[1], Disha Sharma [1], Trieu Nguyen[1], Ramendra Kundu[1], Michael P Fischbein[5,6], Jesse Engreitz [4], Anshul B Kundaje [3,4], Paul P Cheng [1,5] & Thomas Quertermous [1,5✉]

## Abstract

**Vascular sites have distinct susceptibility to atherosclerosis and aneurysm, yet the epigenomic and transcriptomic underpinning of vascular site-specific disease risk is largely unknown. Here, we performed single-cell chromatin accessibility (scATACseq) and gene expression profiling (scRNAseq) of mouse vascular tissue from three vascular sites. Through interrogation of epigenomic enhancers and gene regulatory networks, we discovered key regulatory enhancers to not only be cell type, but vascular site-specific. We identified epigenetic markers of embryonic origin including developmental transcription factors such as *Tbx20*, *Hand2*, *Gata4*, and *Hoxb* family members and discovered transcription factor motif accessibility to be vascular site-specific for smooth muscle, fibroblasts, and endothelial cells. We further integrated genome-wide association data for aortic dimension, and using a deep learning model to predict variant effect on chromatin accessibility, ChromBPNet, we predicted variant effects across cell type and vascular site of origin, revealing genomic regions enriched for specific TF motif footprints—including MEF2A, SMAD3, and HAND2. This work supports a paradigm that cell type and vascular site-specific enhancers govern complex genetic drivers of disease risk.**

**Keywords** Epigenomics; Genomics; Vascular Biology; Single-Cell Transcriptomics; Development
**Subject Categories** Chromatin, Transcription & Genomics; Vascular Biology & Angiogenesis

## Introduction

The risk to develop vascular disease is vascular site-specific—a clinical observation that has been well described for over a half century (Haimovici and Maier, 1964; Haimovici et al, 1958, 1959), yet the biological mechanisms by which this occurs remain poorly understood. Smooth muscle cells (SMC), fibroblasts, and endothelial cells, which make up vascular tissues, have distinct developmental origins (Katz et al, 2012; Majesky, 2007; Mikawa and Gourdie, 1996; Red-Horse et al, 2010; Sawada et al, 2017; Tian et al, 2013; Volz et al, 2015) and this embryonic lineage diversity has been hypothesized to contribute to vascular site specific susceptibility to disease (Majesky, 2007). Genome-wide association studies (GWAS) of vascular disease have revealed unique loci for disease in individual vascular sites, highlighting differential genetic and molecular etiologies for disease in differing vascular tissues (Franceschini et al, 2018; Jones et al, 2017; Nikpay et al, 2015; Pirruccello et al, 2022; Schunkert et al, 2011; Strawbridge et al, 2020; van der Harst and Verweij, 2018). Understanding the fundamental mechanisms that mediate this differential susceptibility to disease is critical to understand mechanisms of vascular disease and discover novel therapeutic targets.

Genomic structure and chromatin accessibility within the cell is intricately linked to function and governs transcriptional programs that drive disease risk (Oudelaar and Higgs, 2021). This context is important when considering that the heritability of complex disease is determined by common genetic variation that largely mediates disease risk through modifying genomic enhancer function within regions of open chromatin (Anene-Nzelu et al, 2022). The recent advancement of single cell global chromatin accessibility profiling through *A*ssay for *T*ransposase *A*ccessible *C*hromatin *seq*uencing (scATACseq) in combination with single cell transcriptomic analysis (scRNAseq) has further defined that genetic risk of cardiac (Alexanian et al, 2021; Hocker et al, 2021) and vascular disease (Cheng et al, 2022; Depuydt et al, 2020; Wang et al, 2021) is driven in part through modification of cell-type-specific enhancers regulating expression of disease-modifying genes in a cell type-specific manner.

During vasculogenesis in early life, the carotid arteries and great vessels of the heart form from the pharyngeal arch arteries which appear in a craniocaudal sequence and then regress or remodel to form a definitive vascular pattern (Anderson et al, 2008; Aquino

[1]Department of Medicine, Division of Cardiovascular Medicine, Stanford University, Stanford, CA, USA. [2]Stanford Center for Inherited Cardiovascular Disease, Stanford, CA, USA. [3]Department of Computer Science, Stanford University, Stanford, CA, USA. [4]Department of Genetics, Stanford University, Stanford, CA, USA. [5]Stanford Cardiovascular Institute, Stanford University, Stanford, CA, USA. [6]Department of Cardiothoracic Surgery, Stanford University, Stanford, CA, USA. ✉E-mail: weldyc@stanford.edu; tomq1@stanford.edu

et al, 2021; Li et al, 2012; Paffett-Lugassy et al, 2013). The adult murine aorta is composed of vascular cells from diverse embryonic origin including the secondary heart field, neural crest, and somitic mesoderm (Majesky, 2007) and reside in spatially distinct domains (Sawada et al, 2017). Further, analysis of smooth muscle cells from adult healthy vascular tissue in mouse has shown a differential gene expression profile across vascular sites, which suggests relevance to vascular site-specific disease risk (Dobnikar et al, 2018; Yu et al, 2022).

The epigenomic landscape of tissues may be influenced by embryonic origin, yet cell-type-specific enhancers of vascular tissues between vascular sites have yet to be mapped. By performing scATACseq combined with scRNAseq of healthy adult mouse aortic tissue from three vascular sites, (1) aortic root and ascending aorta, (2) brachiocephalic and carotid artery, and (3) descending thoracic aorta—representing the secondary heart field and neural crest, neural crest, and somitic mesoderm, respectively—we report that cell-type-specific enhancers of vascular cells are vascular site specific, suggest a developmental "memory", and are related to disease risk genes. Through a comprehensive approach utilizing gene regulatory networks (GRN), in vitro culture of primary adventitial fibroblasts, and a novel technique to develop machine learning neural networks (ChromBPNet) to predict human variant effect within cell types across vascular sites, we not only reveal key differential chromatin accessibility of transcription factor (TF) motifs between vascular sites but provide data to support the hypothesis that genetic variants regulate cell function in not only a cell type, but vascular site-specific mechanism. This work provides important insights into disease risk across vascular tissues and supports the concept that the epigenomic landscape of vascular cells is vascular site-specific and regulates disease risk.

# Results

## Single-cell analysis of vascular tissue reveals cell type and vascular site-specific epigenomic profiles

We performed microdissections to collect the vascular tissues from three vascular sites in 16 healthy adult C57Bl/6 male mice (14–16 weeks of age) including the (1) ascending aorta and aortic root, (2) brachiocephalic and right common carotid arteries, and (3) descending thoracic aorta (Fig. 1A). Tissues were pooled by vascular site, underwent enzymatic digestion and mechanical dissociation, live cells were FACS sorted, and cells were then partitioned for either scRNAseq or for subsequent nuclei isolation and scATACseq using 10X genomics platform as previously described (Cheng et al, 2022; Kim et al, 2020; Wirka et al, 2019). Following quality control and removal of low-quality cells, data were normalized and underwent linear dimensional reduction following standard protocols in R packages Seurat and Signac (Stuart et al, 2019). A total of 37,394 cells and 40,275 cells were of high data quality and were profiled with scRNAseq and scATACseq, respectively, with roughly 10,000–15,000 cells per vascular site and sequencing modality. Visualization with UMAP methodology for both scRNAseq and scATACseq datasets was performed (Fig. 1B,C). UMAP visualization of scRNAseq and scATACseq data reveals a specific pattern where vascular site origin distinguishes cell clusters (Fig. 1B,C).

Given the interrelationship between RNA expression and global chromatin accessibility, these datasets were then integrated together as previously described (Cheng et al, 2022; Stuart et al, 2019). Integrated scRNAseq and scATACseq datasets underwent additional non-linear dimensional reduction and UMAP visualization (Fig. 1D). The integrated UMAP representing both transcriptomic and chromatin accessibility visually reveals significant differences between cell types by vascular site (Fig. 1D). Analysis of RNA expression of canonical genes to differentiate cell type shows strong correlation to overall gene chromatin activity (e.g., *Myh11*—smooth muscle, Fig. 1E,F). Integrated cell clusters were then assigned to the major cell types of the vascular wall, vascular smooth muscle cells (SMCs), fibroblasts, endothelial cells, and macrophages (Fig. 1G) using known lineage markers. To further focus on SMC and fibroblast cell populations, a subset analysis of SMC and fibroblasts was performed with UMAP visualization of scRNAseq, scATAC-seq, and integrated datasets, demonstrating clear separation of cells by original vascular site, demonstrating a unique vascular site-specific chromatin and RNA profile by cell type (Fig. 1H,I).

## Epigenomic patterns of vascular SMCs are cell type and vascular site-specific

UMAP clustering of SMCs from RNA, ATAC, and integrated datasets demonstrates a pattern where SMCs from the ascending aorta, carotid artery, and descending thoracic vascular sites are distinct from one another (Fig. 1H). Chromatin peak accessibility analysis between ascending and descending thoracic aorta SMCs identifies 4805 peaks which are differentially accessible (Fig. 2A). Peak locations were analyzed using Genomic Regions Enrichment of Annotations Tool (GREAT) (McLean et al, 2010). GREAT performs genomic region–gene associations by assigning a regulatory domain for each gene, and then each genomic region is associated with all genes whose regulatory domain it overlaps. Of these peaks, the majority (3971/4805; 83%) are associated with two genes within a region (Fig. 2B; Dataset EV1). A distribution of region–gene association distances to TSS is observed where most peaks lie downstream of the TSS (Fig. 2C). Roughly 12% of region–gene associations are within 5 kb of the TSS of a gene (Fig. 2C) and ~43% (4496/8789) of region–gene associations are within 50 kb of a TSS (Fig. 2C).

When comparing peak accessibility between ascending and descending aorta, specific chromatin regions are identified as having marked differential accessibility. Among the top differentially accessible peaks in the ascending SMC population lie at chr8—57,320,451–57,321,235 and chr14—63,244,897–63,245,811, 140 bp and 83 bp upstream from the TSSs for *Hand2* and *Gata4* (Fig. 2D,E). Coverage plots of chromatin accessibility reveal *Hand2* regulatory regions to be open in ascending and carotid SMCs while closed in descending SMCs (Fig. 2D). This contrasts with regulatory regions of *Gata4* that appear to be in an open state in the ascending SMC however in a largely closed state in the carotid and descending SMCs (Fig. 2E). Consistent with the role of Hox genes during the development of somites, top peaks that have increased accessibility in the descending aortic SMC include genomic regions that lie near the *Hoxb* family of transcription factors (i.e., chr11—9,686,379–96,287,301 at *Hoxb7*, Fig. 2F; Dataset EV1). Feature plots of peak accessibility for *Hand2* and *Hoxb7* peaks (Fig. 2G–I) highlight distinct differential accessibility

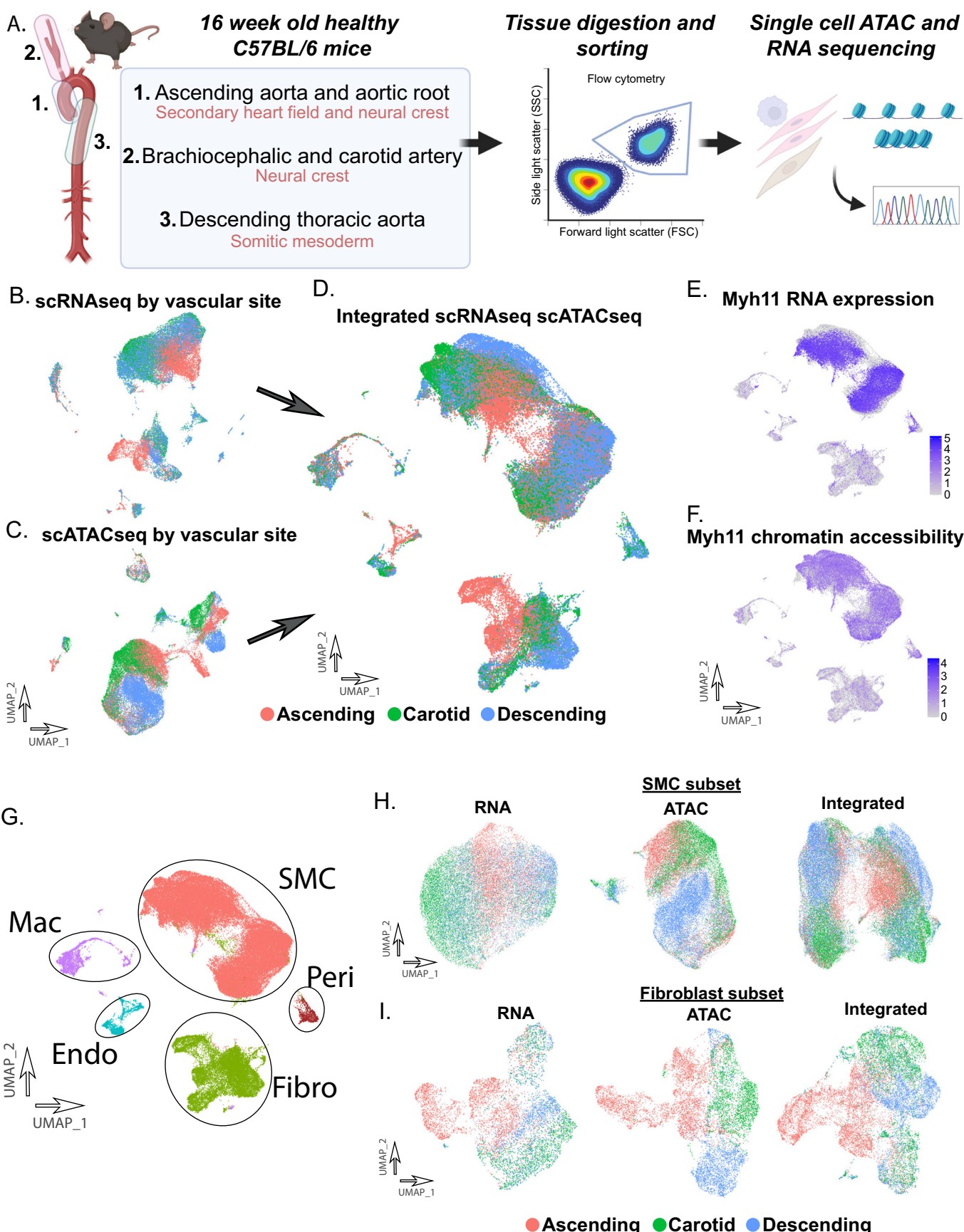

**Figure 1.   Transcriptomic and epigenomic landscape of single vascular cells.**

Single-cell RNA seq (scRNAseq) and single-cell ATAC seq (scATACseq) was performed on vascular tissue in adult healthy mice C57Bl/6) in three vascular sites (aortic root/ascending aorta, brachiocephalic/carotid artery, descending thoracic aorta) (**A**). UMAP of scRNAseq data (**B**), scATACseq data (**C**), and integrated datasets (**D**). RNA expression *Myh11* in the integrated dataset (**E**). Chromatin accessibility of *Myh11* in the integrated dataset (**F**). Differentiation of major cell populations of the vascular wall (**G**). Subset analysis for scRNA, scATAC, and integrated datasets in SMC (**H**) and Fibroblast (**I**).

dependent on SMC vascular site of origin. Consistent with epigenetic marks, RNA expression analysis reveals *Hand2* expression in the ascending and carotid SMC while absent expression in the descending SMC (Fig. 2J). Top differential peak analysis identifies other cardiac and vascular development genes that have higher chromatin accessibility within the ascending aorta include *Tbx20*, *Tbx2*, *Gata4*, and *Wnt16*, while homeobox genes such as the *Hoxa/b/c* family of genes, *Smad2*, and are identified as having higher chromatin accessibility within the descending aorta (Datasets EV2 and 3).

Evaluation of biological processes of differentially accessible chromatin regions using GREAT identifies key development pathways that characterize the ascending aorta SMC such as "outflow tract morphogenesis", "cardiac septum development", and "embryonic heart tube development" as well as other pathways involving cellular signaling such as "transmembrane receptor protein signaling", "regulation of Ras protein" (Appendix Fig. S1A). Biological processes that characterize the descending aorta SMC include developmental programs such as "anterior/posterior pattern specification", however notable processes regulating cytoskeletal organization such as "actin cytoskeleton organization", "actomyosin structure organization", and "regulation of SMC differentiation", as well as processes involved in the negative regulation of TGFβ signaling such as "negative regulation of TGFβ receptor signaling" (Appendix Fig. S1B).

We then performed transcription factor motif accessibility analysis using ChromVAR as previously described (Schep et al, 2017). In comparing ascending versus descending aorta SMCs, we identified that ascending SMCs have notable transcription factor motif enrichment for AP1 factors (i.e., FOS:JUNB "TGAGTCA", Fig. 2K), neural crest and secondary heart field transcription factors such as *HAND2*, *GATA4*, and *TCF21* (i.e., *HAND2*, Fig. 2L), as well as TFs with known roles in SMC differentiation such as *TWIST1* and *TEAD* factors (Dataset EV4). Top differentially accessible motifs in the descending SMCs highlight GC-rich KLF motifs (i.e., *KLF15*, Fig. 2M), numerous AT-rich homeobox transcription factor motifs, including HOX factors/Cdx1 (caudal type homeobox 1)/ Lhx1 (LIM homeobox 1), and MEF2 factors (i.e., *MEF2A*, Fig. 2N; Dataset EV5). Motif enrichment analysis of the carotid SMCs suggests similarities to ascending SMCs with notable enrichment for AP1 and *HAND2* motifs (Dataset EV6), however, we identified additional motif enrichment that appeared to be largely distinct to the carotid SMC, including distal-less homeobox (DLX) motifs (*DLX5*, Fig. 2O) as well as neurodevelopmental E-box TFs such as *NEUROD1* and *NEUROG2* (*NEUROD1*, Fig. 2P).

Gene expression analysis using FindMarker from scRNAseq data of SMCs reveals 252 genes which have differential expression among vascular sites (147 ascending, 77 carotid, 28 descending) (Dataset EV7) that largely match genes identified by differential peak accessibility analysis (Fig. 2Q). Top differentially expressed genes within the ascending SMC population includes expected

developmental genes such as *Tbx20* and *Hand2* (Fig. 2Q), secondary heart field marker *Tnnt2* (Fig. 2Q,R), but also numerous genes with previously demonstrated roles in atherosclerosis suggesting vascular site-specific disease risk mechanisms (Fig. 2Q), including *Ccn3* (cellular communication network factor 3) which has previously reported vascular protective effects by inhibiting neointimal formation and plaque development (Liu et al, 2014; Shimoyama et al, 2010) and *Dcn* (decorin) that has been shown to be protective in atherosclerosis (Al Haj Zen et al, 2006). Notably, we identify specific vascular development and inflammation genes as having higher expression within SMCs isolated from the carotid artery, such as *Klf4*, *Fosb*, *Jun*, and *Atf3* (Fig. 2Q,R). *Fosb* and *Jun* have specific roles in mediating cellular and arterial contractility (Licht et al, 2010), *Atf3* is a CAD GWAS gene and has recently been identified to have vascular protective effect in atherosclerosis (Wang et al, 2021), and *Klf4* has critical roles in mediating SMC phenotypic modulation and promotes atherosclerosis (Yoshida et al, 2008). Specific genes that have higher expression within the descending thoracic SMCs include signal transduction genes such as *Rgs5* (Regulator of G protein signaling 5) (Fig. 2Q; Dataset EV7), *Fn1* (fibronectin 1), and *Ccdc3* (coiled-coil domain containing 3). *Rgs5* has been reported to be a specific marker of peripheral arterial smooth muscle, is downregulated in atherosclerosis, and acts to inhibit SMC proliferation and neointimal formation (Daniel et al, 2016; Li et al, 2004), *Fn1* is a putative CAD GWAS gene thought to be protective in CAD and loss of function has been implicated in thoracic aortic aneurysm (Paloschi et al, 2011; Soubeyrand et al, 2022), and *Ccdc3* has been found to inhibit TNFα mediated vascular inflammation (Azad et al, 2014).

## Fibroblast cell subset analysis reveals vascular site-specific gene programs and disease risk genes

Evaluation of fibroblast subset population similarly reveals differences in chromatin accessibility and gene expression across vascular sites (Fig. 1I). Peak accessibility analysis between fibroblasts from the ascending and descending aorta reveals 7008 peaks that are differentially accessible (Fig. 3A; Dataset EV8). Top peaks that have higher accessibility in the ascending aorta include peaks located at chr9—24,773,921–24,774,837 and chr14—63,244,897–63,245,811 that lie 95 bp and 85 bp upstream from the TSSs of *Tbx20* and *Gata4* respectively, while peak chr15—57,985,530–57,986,444 is increased in both ascending and carotid and lies 568 bp downstream from the TSS of *Fam83a* (Fig. 3B–D). Top peaks with higher accessibility in the descending aorta fibroblast population includes peaks near Hox related family members (i.e., chr11—96,298,662–96,299,576 at *Hoxb6*) (Fig. 3E), however additional peaks include chr17—35,590,469–35,591,345 in a gene desert 23,457 bp downstream of noncoding RNA 2300002M23Rik, as well as chr8—114,679,333–114,680,184 that lies within an intron and 240,104 bp downstream of the TSS of

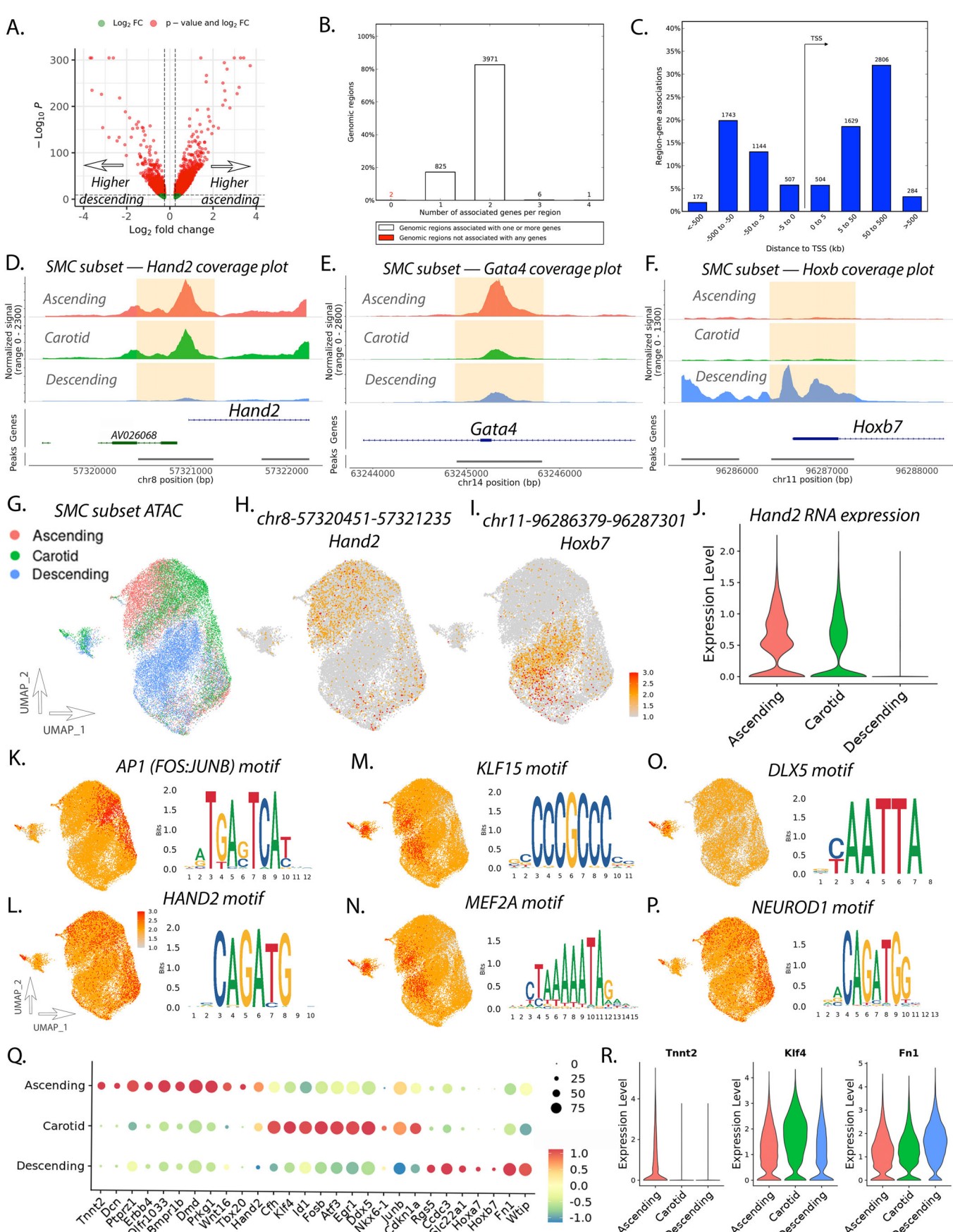

◄  **Figure 2.   SMC subset analysis reveals differential chromatin peak accessibility, gene expression, and transcription factor motif accessibility between vascular sites.**

Volcano plot of differentially accessible peaks comparing ascending and descending SMC (**A**). Peak to gene analysis revealing the number of associated genes per genomic region (**B**). Histogram of distance from peak to associated gene TSSs (**C**). Coverage plots of *Hand2* (**D**), *Gata4* (**E**), and *Hoxb* (**F**) for ascending, carotid, and descending SMC. UMAP of scATACseq data for SMC subset (**G**). Feature plot of *Hand2* (**H**) and *Hoxb7* (**I**) peaks. Violin plot of *Hand2* RNA expression between vascular sites (**J**). Feature plot of motif accessibility and motif sequence for *AP1* (**K**), *Hand2* (**L**), *Klf15* (**M**), *Mef2a* (**N**), *Dlx5* (**O**), and *Neurod1* (**P**). Dotplot of RNA expression for top differentially expressed transcripts (**Q**). Violin plot of SMC RNA expression across vascular sites for *Tnnt2*, *Klf4*, and *Fn1* (**R**). Differential peak accessibility, RNA transcript expression, and TF motif accessibility analysis between groups was performed with the Wilcoxon rank-sum test. SMC subset analysis represents 25,665 cells for RNA and 28,784 cells for ATAC across vascular sites.

*Wwox* (Fig. 3F,G). Feature plots of peak accessibility to peaks that correspond to *Fam83a* and *Wwox* (chr15—57,985,530–57,986,444 and chr8—114,679,333–114,680,184, respectively) highlight distinct vascular site-specific chromatin accessibility (Fig. 3H,I).

Peak to gene association analysis with GREAT reveals top genes with higher peak accessibility in the ascending aorta fibroblasts to include expected key developmental genes such as *Tbx20*, *Gata4*, *Tcf21*, and *Hand2*, as well as signaling genes such as *Fam83a*, *Cul4b*, and *Cdkn2b* (Dataset EV9). Peaks with higher accessibility within the descending aorta fibroblasts include peaks that lie near the *Hoxa/b/c* family of transcription factors as well as other developmental transcription factors such as *Pax1* and *Foxd1* (Dataset EV10). Gene ontology pathway analysis using GREAT identifies top pathways to be enriched in the ascending aorta to include developmental pathways such as "cardiac septum morphogenesis" and "embryonic heart tube development", but also pathways that suggest a distinct response to TGFβ such as "regulation of TGFβ receptor signaling" and "regulation of cellular response to TGFβ stimulus" (Appendix Fig. S2A). Pathways enriched within the descending aorta fibroblast include an enrichment of pathways involved in extracellular structure organization such as "extracellular matrix organization", "collagen fibril organization", and "negative regulation of cell–cell adhesion" (Appendix Fig. S2B).

Motif accessibility analysis using ChromVar within fibroblast populations reveals ascending fibroblasts to have a marked increase in accessibility of *TCF21* and other bHLH family of transcription factors, similar to what was observed in the SMC ascending population (Fig. 3J,K; Dataset EV11). However, motif enrichment appears to further identify regulators of TGFβ signaling including the SMAD2:3:4 and TGIF1 motifs (Fig. 3L). *SMAD3* is a member of the TGFβ superfamily and has causal roles in aneurysm (van de Laar et al, 2011) and coronary artery disease (Iyer et al, 2018). Like that observed in the SMC, descending aortic fibroblast appear to have increased accessibility of motifs that relate to homeobox transcription factors such as HOX family of TFs (i.e., *HOXA13*, Fig. 3M), *MEIS1/2*, GC-rich KLF family members, as well as *SOX9* and other SOX TFs, which further appear enriched in the descending fibroblasts (Fig. 3N; Dataset EV12). Motif enrichment for carotid fibroblast appears to show a distinct increase in accessibility to developmental TF *NKX6-1* (Fig. 3O) as well as related TFs with similar sequence (i.e., *MEOX1/2*) (Dataset EV13).

RNA expression analysis between vascular sites demonstrates significant concordance with peak accessibility analysis. Fibroblast-specific gene expression analysis reveals 597 genes which have differential expression between vascular sites (314 ascending, 149 carotid, 134 descending) (Fig. 3P; Dataset EV14). The top 10 genes which differentiate vascular sites include expected development

genes such as *Tbx20*, *Tcf21* (ascending), *Igfbp4*, *Sod3* (carotid), and *Hoxa7*, *Col1a2* (descending) (Fig. 3Q). A notable observation is that descending fibroblasts appear to be enriched for extracellular structure organization pathways by scATACseq (Appendix Fig. S2B) and by RNA we observe a pattern of increased expression of collagen extracellular matrix genes (Fig. 3Q,R). These genes overlap with known causal genes for hereditary thoracic aortopathy (HTAD) and loci associated with aortic dimension and aortic dissection risk (Chou et al, 2023; Renard et al, 2018). Evaluation of these genes reveal a pattern where TGFβ-related HTAD genes in which gain of function leads to disease (i.e., *TGFBR1/2*, *TGFB2*, *SMAD3*) show increased expression in the ascending fibroblasts, while collagen and matrix-related HTAD and other collagen genes, where loss of function leads to disease (i.e., *FBN1*, *COL3A1*, *LOX*) there is decreased expression (Fig. 3R). This is notably relevant to Loeys-Dietz syndrome, characterized by abnormal TGFβ signaling that preferentially results in thoracic aortic disease. This suggests distinct regulation of extracellular matrix and TGFβ signaling genes in fibroblasts in a vascular site-specific manner that may suggest a relationship to the risk of thoracic aortic disease.

## Endothelial cell subset analysis shows distinct vascular site-specific chromatin accessibility

Single-cell analysis of endothelial cell subset population reveals vascular site-specific chromatin accessibility and gene expression programs. Visualization with UMAP demonstrates separation of endothelial cells from the ascending aorta with somewhat less pronounced differences between endothelial cells from the carotid artery and descending aorta based on RNA, ATAC, and integrated datasets (Fig. 4A–C). Peak accessibility analysis between endothelial cells from the ascending and descending aorta identifies 935 peaks with differential accessibility meeting significance using an adjusted *P* value of 0.05 (Fig. 4D) with 14,388 peaks meeting an unadjusted *P* value of 0.05 (Dataset EV15). Top peaks that have increased accessibility in the ascending aorta include peak chr6—134,981,212–134,982,145 which lies 39 bp upstream of the TSS of *Apold1* (Fig. 4E,F). *Apold1* (apolipoprotein L domain containing 1, aka VERGE) has been previously reported as an endothelial cell-specific stress response gene and protects against vascular thrombosis (Diaz-Canestro et al, 2020; Regard et al, 2004). Peak analysis further highlights peaks related to Wnt signaling, including peak chr6—18,031,351–18,032,250, which lies 1216 bp upstream of the TSS for *Wnt2* (Fig. 4G). Coverage plot of this genomic region demonstrates that chromatin accessibility is nearly entirely closed in the descending aorta with only minor accessibility in the carotid (Fig. 4G). Notably, the descending aorta endothelial cells have increased accessibility of Hox related genes (i.e., peak chr6—

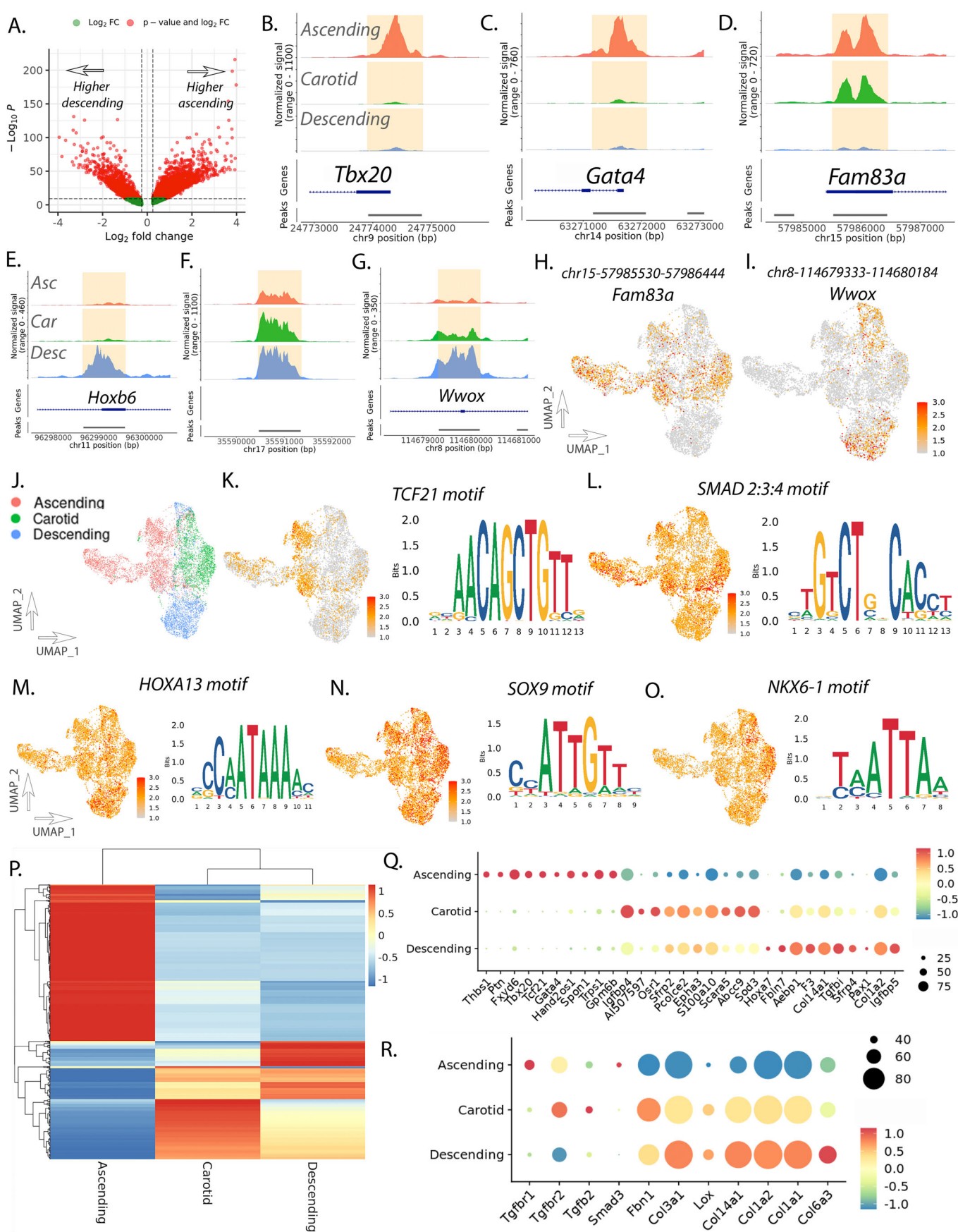

**Figure 3. Fibroblast subset analysis reveals vascular site-specific enhancers for developmental and disease genes.**

Volcano plot of differentially accessible peaks between ascending and descending aortic fibroblasts (**A**). Coverage plot for ascending, carotid, and descending aortic fibroblasts at *Tbx20* (**B**), *Gata4* (**C**), *Fam83a* (**D**), *Hoxb6* (**E**), *chr17* gene desert (**F**), and *Wwox* (**G**). Feature plot of scATACseq data of fibroblast subset for peaks near *Fam83a* (**H**) and *Wwox* (**I**). UMAP of scATACseq data from fibroblast subset by vascular site (**J**). Feature plot of scATACseq for motif accessibility and motif sequence for *Tcf21* (**K**), *Smad2:3:4* (**L**), *Hoxa13* (**M**), *Sox9* (**N**), and *Nkx6-1* (**O**). Heatmap of scRNAseq differentially expressed genes in ascending, carotid, and descending fibroblast subset (**P**). Dotplot of scRNAseq transcript expression for top differentially expressed genes in fibroblast subset (**Q**). Dotplot of scRNAseq transcript expression for hereditary aortopathy genes (**R**). Differential peak accessibility, RNA transcript expression, and TF motif accessibility analysis between groups was performed with the Wilcoxon rank-sum test. Fibroblast subset analysis represents 7586 cells for RNA and 8973 cells for ATAC across vascular sites.

52,225,707–52,226,553, *Hoxa9*) (Fig. 4H,I), as well as peaks related to genes involved in Notch signaling such as chr2—137,109,516–137,110,365 which lies 6707 bp downstream of the TSS for *Jag1* (Fig. 4J) where coverage plot of this genomic region reveals differential peak accessibility with carotid and descending endothelial cells having similar accessibility (Fig. 4J). *Jag1* (jagged 1) and associated Notch pathway signaling has important roles in suppressing vascular smooth muscle chondrogenic fate and in the formation of the atherosclerotic fibrous cap (Briot et al, 2014; Martos-Rodriguez et al, 2021).

Motif accessibility analysis was performed comparing ascending and descending aortic endothelial cell populations. Top differentially accessible motifs in the ascending endothelial cells highlight *LEF1* (Fig. 4K; Dataset EV16). *LEF1* plays a critical role in Wnt β-catenin signaling and regulates endothelial cell fate specification (Hubner et al, 2017). Other motifs enriched in the ascending population include *Hand2* as well as multiple ETS factors (Dataset EV16). ETS factors play important roles in cellular growth and differentiation and regulate vascular inflammation and remodeling where inhibition of ETS transcription factors promotes vessel regression (Oettgen, 2006; Schafer et al, 2020). Top motifs enriched in the descending aortic endothelial cells highlight *NFIC* (Fig. 4L; Dataset EV17) where motif accessibility appears to be enriched in both the carotid and descending endothelial cells. Other motifs include multiple Hox family members, TEAD family members, and other nuclear factors such as *NFIX* (Dataset EV17). Motif enrichment in carotid endothelial cells highlights nuclear factors such as *NFIA/C/X* as well as AP1 factors (Dataset EV18).

Differential gene expression analysis reveals specific patterns differentiating endothelial cell programs by vascular site (Fig. 4M) with FindMarker analysis identifying 397 differentially expressed genes between vascular sites (174 ascending, 127 carotid, 96 descending) (Dataset EV19), where heatmap analysis suggests greater similarity between carotid and descending endothelial cells compared to ascending (Fig. 4M). Ascending aortic endothelial cells have increased expression of VEGF receptors *Kdr* (kinase insert domain receptor, aka VEGFR), *Flt1* (fms related receptor tyrosine kinase 1, aka VEGFR1), and *Flt4* (fms related receptor tyrosine kinase 4, aka VEGFR3) (Fig. 4N,O). Descending aortic endothelial cells have notable increase in expression of *Edn1* as well as other BMP factors (i.e., *Bmp4*) as well as interestingly xenobiotic transformation gene *Cyp1b1* (cytochrome p450, 1b1) (Fig. 4P,Q). Vascular site-specific expression of *Cyp1b1* is particularly notable as it contributes to abdominal aortic aneurysm and suggests a vascular site-specific mechanism of aneurysm (Mukherjee et al, 2021; Thirunavukkarasu et al, 2016). Carotid endothelial cells have higher expression of specific genes that appear to play unique roles within the cerebral vasculature such as *Efemp1* (EGF containing

fibulin extracellular matrix protein 1) (Fig. 4R) an extracellular matrix glycoprotein which regulates vessel development and has been associated with intracranial vascular disease and white matter density through large genome-wide association studies (Traylor et al, 2019).

## Macrophage cells have minimal epigenomic and transcriptional variation across vascular sites

Given the significant vascular site-specific epigenomic and RNA transcriptional profiles observed within vascular SMCs, fibroblasts, and endothelial cells, we next evaluated if resident macrophages within healthy vascular tissues harbor specific epigenomic and transcriptional profiles. Macrophage cells from integrated scATACseq/scRNAseq data were subsetted and subsequently analyzed. Importantly, UMAP visualization of macrophage cells with RNA, ATAC, and integrated datasets reveals no significant differences in macrophage cells across vascular sites (Appendix Fig. S3A–C). Differential peak accessibility analysis between ascending and descending aorta macrophages do not reveal any peaks which meet statistical significance as differentially accessible (Appendix Fig. S3D). Differential gene expression analysis with scRNAseq data reveal very minimal differences in gene expression (Appendix Fig. S3E). This lack of vascular site-specific macrophage diversity across the aorta is notable given the contrasting finding within SMCs, fibroblasts, and endothelial cells, and that significant leukocyte diversity develops within the aorta during atherosclerosis (Zernecke et al, 2020).

## Cell type and vascular site-specific gene regulatory networks highlight distinct ascending fibroblast regulatory networks

These data indicate that the epigenomic landscape is not only cell type but vascular site-specific, with key differences in transcription factor motif accessibility. To further understand how TF regulatory elements may further control gene expression on a cell type and vascular site-specific basis, we utilized this multi-modal single-cell data to infer gene regulatory networks (GRNs) with Pando (Fleck et al, 2023). By integrating scRNA and scATAC datasets and subsetting across vascular cell type and vascular site of origin, we utilized Pando to infer GRNs through modeling gene expression with understanding the interaction of TF expression with TF-binding sites and gene targets (Fig. 5A). With Pando, we initiated GRN analysis by scanning candidate genomic regions and identified TF-binding motifs to then infer GRN modules. To understand how regulatory networks may differ within cell type based on vascular site, we identified TF modules for aortic SMCs

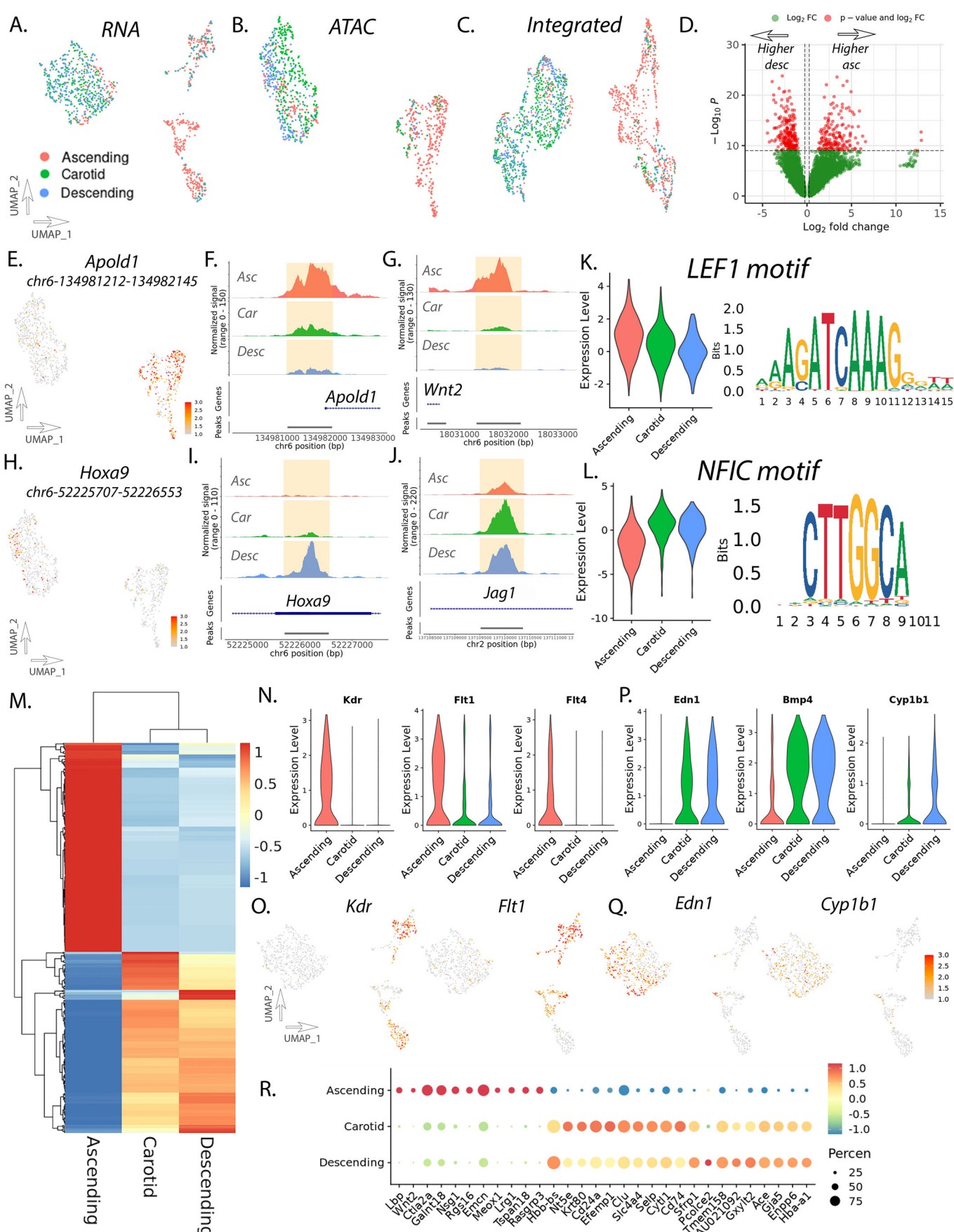

◀ **Figure 4. Endothelial cell subset analysis identifies differentially accessible chromatin peaks and differential transcription factor (TF) motif accessibility.**

UMAP of RNA (**A**), ATAC (**B**), and integrated (**C**) datasets for the endothelial cell subset. Volcano plot of differentially accessible peaks between ascending and descending endothelial cells (**D**). Feature plot of peak accessibility of *Apold1* peak (**E**). Coverage plots for *Apold1* (**F**) and *Wnt2* (**G**) genomic regions in the endothelial cell subset. Feature plot of peak accessibility of *Hoxa9* peak (**H**). Coverage plots for *Hoxa9* (**I**) and *Jag1* (**J**) genomic regions in the endothelial cell subset. Violin plot of motif accessibility and motif sequence for *LEF1* (**K**) and *NFIC* (**L**). Heatmap of differential RNA expression by endothelial cell vascular site (**M**). Violin plots of RNA expression in endothelial cells for *Kdr*, *Flt1*, and *Flt4* (**N**). Feature plots of RNA expression for *Kdr* and *Flt1* (**O**). Violin plots of RNA expression in endothelial cells for *Edn1*, *Bmp4*, and *Cyp1b1* (**P**). Feature plots of RNA expression for *Edn1* and *Cyp1b1* (**Q**). Dotplot of scRNAseq transcript expression for top differentially expressed genes in endothelial cell subset (**R**). Differential peak accessibility, RNA transcript expression, and TF motif accessibility analysis between groups was performed with the Wilcoxon rank-sum test. Endothelial subset analysis represents 1085 cells for RNA and 982 cells for ATAC across vascular sites.

and Fibroblasts in both ascending and descending aortic sites. In these modules, we identified GRN TF and TF gene targets. In the SMC population, we identified 37 and 38 TF modules in the ascending and descending cells, respectively. In the fibroblast population, we identified 56 and 40 TF modules in the ascending and descending cells, respectively (Fig. 5B,C). In both SMC and fibroblast cell populations, the majority of TF modules appear to be shared between ascending and descending cells (Fig. 5B,C), with key TFs including *Atf3*, *Creb5*, *Fosb*, *Klf2/4/6*. However, although SMC TF modules show a relatively equal proportion of TF modules that are distinct to ascending and descending, ascending fibroblasts appear to have a notable increase in distinct TF modules (20 vs 4) (Fig. 5C). Distinct ascending fibroblast GRN TF modules include *Meox1*, which has recently been described as a master regulator of fibroblast activation in cardiac fibroblasts through epigenetic mechanisms (Alexanian et al, 2021). GRN TF gene targets appear to have less overlap between ascending and descending cell populations in both SMCs and fibroblasts (Fig. 5D,E), however consistent with GRN TF modules, there is a greater proportion of ascending fibroblast GRN gene targets compared to descending (108 vs 43) (Fig. 5D,E). Visualization of ascending fibroblast GRN highlights the proximity of *Meox1* to other developmental TFs *Tbx20* and *Gata4* (Fig. 5F). Other distinct fibroblast ascending GRN TF modules include *Tcf21*, *Irf7/8*, *Sox4/7/17*, and *E2f8* (Appendix Fig. S4).

### Vascular site-specific Meox1 activation implicates epigenetic 'priming' for fibroblast activation in ascending aortic fibroblasts

Our findings implicate a cell type and vascular site-specific 'epigenetic memory' of important regulatory enhancer elements with functional effect on disease-relevant pathways. A notable observation from the evaluation of GRNs is that ascending fibroblasts have increased proportion of distinct regulatory networks from descending fibroblasts highlighting the TF *Meox1*. We interrogated our scRNAseq and scATACseq fibroblast integrated dataset where we see that *Meox1* has increased RNA expression in ascending fibroblasts, but this expression is largely limited to a specific cell population that represents valvular fibroblasts however also includes other ascending aorta adventitial fibroblast populations (Fig. 5G,H). Importantly, we see that *Meox1* chromatin accessibility is much more expansive and extends throughout all ascending and includes carotid fibroblasts, with increased total chromatin accessibility in the ascending/carotid fibroblast populations (Fig. 5I). This discrepancy of chromatin accessibility and

RNA expression for non-valvular ascending fibroblasts may suggest evidence to support an epigenetic "priming".

### Primary adventitial fibroblast response to TGFβ is dependent on vascular site and implicates functional developmental epigenomic memory

We have shown that vascular SMCs, fibroblasts, and endothelial cells have transcriptional and epigenomic features that are distinct to the vascular site. Further, TF motif accessibility analysis in fibroblasts reveals an increased accessibility of *AP1*, and *SMAD2:3:4*, and *TCF21* motifs in the ascending fibroblast population compared to carotid and descending fibroblasts (Fig. 3), with further gene regulatory network analysis suggesting distinct regulatory TF module activity in ascending fibroblasts highlighting master regulator of fibroblast activation *Meox1* (Fig. 5). This differential chromatin accessibility would suggest the potential for heightened biological response to TGFβ in ascending fibroblasts. To evaluate the functional effect of this differential chromatin accessibility and to identify if differential gene expression is retained following removal from vascular site flow conditions, we isolated and cultured primary adventitial fibroblasts from healthy 14 week old C57BL/6 mice from the ascending and descending aorta, passaged cells 3X allowing for separation from hemodynamic effects, and stimulated them with control or TGFβ (10 ng/mL, 48 h) and performed bulk RNA sequencing ($n = 3$ per condition) (Fig. 6A).

Principal component analysis (PCA) of RNAseq data reveals distinct separation of samples based on origin of vascular site and stimulation with TGFβ (Fig. 6B). Upon evaluation of differential gene expression by vascular site, we observe 1342 differentially expressed genes between ascending and descending fibroblasts (Fig. 6C; Dataset EV20). Ascending fibroblasts have increased expression of genes including *Hand2*, *Tbx20*, *Pcolce2*, *Tcf21*, and *Gata4* (Fig. 6C; Appendix Fig. S5A), while descending fibroblasts have increased expression of *Hox* family of genes (i.e., *Hoxb9*, *Hoxa7*, *Hoxc8*), *Col6a3*, and *Pax1* (Fig. 6C; Appendix Fig. S5B). When evaluating response to TGFβ stimulation, comparing all samples by TGFβ treatment, we identify 4,234 DE genes, with top upregulated genes including *Ada*, *Snx30*, *Wisp1*, *Cthrc1*, *Loxl2*, and *Akap5*, while top downregulated genes include *Adm*, *Gstm1*, *Mcc*, *Rras2*, *Il6ra*, and *Vegfd* (Fig. 6D; Dataset EV21). To evaluate the effect of TGFβ by vascular site, we compared the transcriptomic effect of TGFβ stimulation within ascending and descending fibroblast samples. We identify TGFβ stimulation to induce a markedly higher transcriptional effect in ascending fibroblasts, with

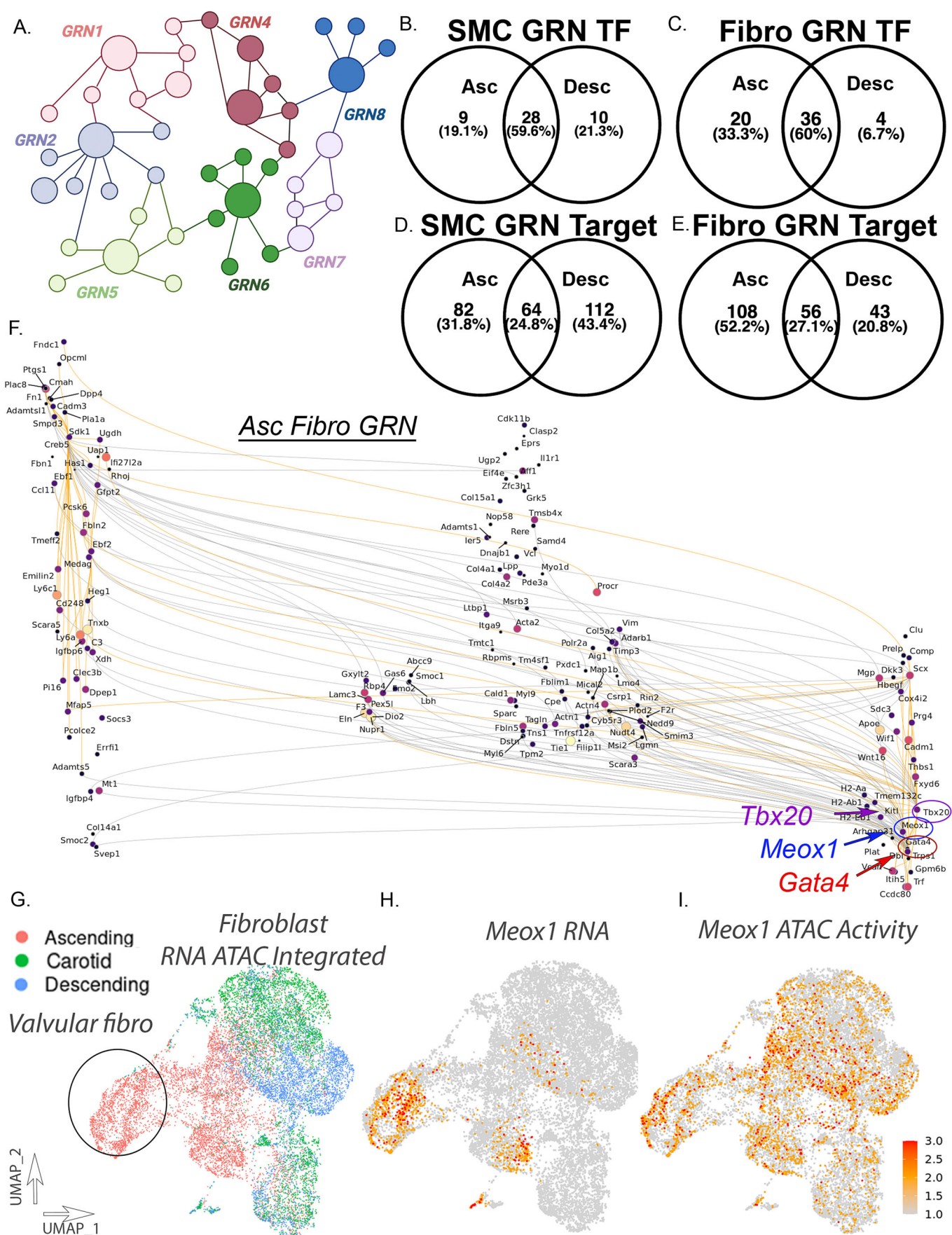

**Figure 5. Gene regulatory network analysis reveals cell type and vascular site-specific regulatory networks within the vessel wall.**

Schematic diagram of gene regulatory networks (GRN) (A). Venn diagrams of GRN transcription factors comparing ascending and descending SMCs (B) and fibroblasts (C). Venn diagrams of GRN transcription factors gene targets comparing ascending and descending SMCs (D) and fibroblasts (E). GRN visualization for ascending fibroblasts network analysis, where each dot represents gene and/or TF, color represents regulatory strength (darker with higher strength), and size of dot reflects the degree of centrality (F). UMAP of scRNA/scATAC integrated fibroblast dataset by vascular site (G). Feature plot of *Meox1* RNA expression (H) and chromatin accessibility (I) in the integrated fibroblast dataset.

6733 DE genes (Fig. 6E; Dataset EV22) while only 2,250 DE genes within descending fibroblasts (Fig. 6F; Dataset EV23).

By performing an interaction analysis, we reveal key genes with differential response to TGFβ by vascular site. In this analysis, 422 genes meet FDR cutoff for interaction significance (Appendix Fig. S6; Dataset EV24). When plotting the log2FC_interaction (log2FC > 0 represents greater effect TGFβ in ascending; log2FC < 0 represents greater effect TGFβ in descending), we identify that matrix gene *Eln* is the top interacting gene that has much lower expression in descending fibroblasts but is rapidly upregulated in response to TGFβ (Appendix Fig. S6A,B). Here, we identify that *Meox1* has a marked interaction in response to TGFβ with greater effect in ascending fibroblasts, where although low expression of *Meox1* is present in ascending fibroblasts, *Meox1* expression significantly increases with TGFβ treatment, an effect not seen in descending fibroblasts (Appendix Fig. S6C). This relationship also includes other TGFβ response genes such as *Gdf6* that are more highly induced in ascending fibroblasts, while *Tgfb1* and *Tgfb3* are more induced in descending fibroblasts (Appendix Fig. S6C; Dataset EV24). Numerous collagen genes have differential responses to TGFβ by vascular site, where *Col8a1* and *Col5a2* are upregulated by TGFβ to a greater extent in descending fibroblasts (Appendix Fig. S6D). Alternatively, *Col28a1* and *Col6a1* are downregulated by TGFβ to a greater extent in descending fibroblasts (Appendix Fig. S6D). TFs such as *Klf4* and *Ptx3* are more highly expressed in ascending fibroblasts and then downregulated by TGFβ to a greater extent than descending fibroblasts (Appendix Fig. S6E). However, *Ahr* is downregulated by TGFβ in descending fibroblasts only while *Hoxb5* shows differing effects by vascular site, where it is modestly upregulated in ascending while downregulated in descending fibroblasts (Appendix Fig. S6E).

By evaluating the 200 bp DNA sequence upstream of the TSS of differentially expressed RNA transcripts in ascending versus descending fibroblast comparison, we evaluated TF motif enrichment using HOMER (Heinz et al, 2010). In control samples, DE genes upregulated in ascending vs descending fibroblasts reveal enrichment of key TFs, including SMAD3 (TWGTCTGV), KLFs including KLF4 (GCCACACCCA), FOS (NDATGASTCAYN), and HIF1α (TACGTGCV) (P values 1.0E-10, 1.0E-13, 1.0E-07, and 1.0E-05, respectively). DE genes upregulated in descending fibroblasts reveal enrichment of TFs, including HOXA11 (TTTTATGGCM), HOXA9 (RGCAATNAAA), HOXC6 (GGCCA-TAAATCA), (P values 1.0E-07, 1.0E-06, 1.0E-03, respectively) (Fig. 6G). We then similarly evaluated the motif enrichment of the 200 bp sequence upstream of the TSS of upregulated genes in ascending fibroblasts in response to TGFβ and then for descending fibroblasts in response to TGFβ (Fig. 6H,I). In this comparison, although motif enrichment to TGFβ has similarities between ascending and descending fibroblasts with GC-rich motifs such as

SP2/5 and KLF factors similarly enriched (Fig. 6H,I), distinct differences emerge. Ascending fibroblast response to TGFβ motif enrichment highlights AP1 factors, SMAD3, as well as MEF2 factors (Fig. 6H). The descending fibroblast response to TGFβ highlights ETS factors and other NRF factors, and to a lesser extent AP1 factors (Fig. 6I). However, SMAD3 and MEF2 factors are not identified as being enriched. The differential enrichment of these TFs in differentially expressed genes between the vascular site and in response to TGFβ suggests this differential chromatin accessibility to have functional effect on the global transcriptome.

## Aortic dimension GWAS identifies vascular site-specific gene set enrichment

Our data indicates that the epigenomic landscape are not only cell type, but vascular site-specific. However, it is unclear if these vascular site-specific enhancers and gene programs relate to human genetic evidence of disease. To evaluate this question, we leveraged the data from a recently performed GWAS to understand the genetic determinants of ascending versus descending aortic dimension (Pirruccello et al, 2022). Through an algorithm-based method to evaluate cardiac MRI imaging of UK Biobank participants, Pirruccello et al, (2022) identified 82 and 47 genomic loci which met genome-wide significance for ascending and descending aortic dimensions, respectively. We hypothesized that if vascular site-specific regulatory enhancers influence genetic disease risk, that we would observe aortic dimension GWAS genes to be differentially expressed in a vascular site-specific manner. By taking the nearest gene for each locus, we evaluated if these GWAS genes are enriched in the ascending versus descending differentially expressed RNA gene lists for each cell type (SMC, Fibro, Endo). Following performing a differential gene expression analysis between ascending and descending aortic SMC, Fibro, and Endo, cell types, we evaluated the proportion of GWAS aortic dimension genes to be differentially expressed. We observed that these GWAS genes are enriched in the differential gene analysis within SMC, Fibroblasts, and Endothelial cells. In SMCs, 25% (21/82) and 21% (10/47) of ascending and descending aortic dimension GWAS genes are differentially expressed (P = 5.4E-6 and P = 2.5E-4, compared to random gene set) (Fig. 7A–C). In Fibroblasts, 32% (26/82) and 34% (16/47) of ascending and descending aortic dimension GWAS genes are differentially expressed (P = 9.9E-5 and P = 1.8E-4, compared to random gene set) (Fig. 7D–F). Whereas, in endothelial cells, 15% (12/82) and 17% (8/47) of ascending and descending aortic dimension GWAS genes are differentially expressed (P = 0.024 and P = 0.019, compared to random gene set) (Fig. 7G–I). These data highlight that a large proportion of genes that regulate aortic dimension are differentially expressed across vascular sites within a cell type and possibly suggest fibroblasts to be the primary cell type.

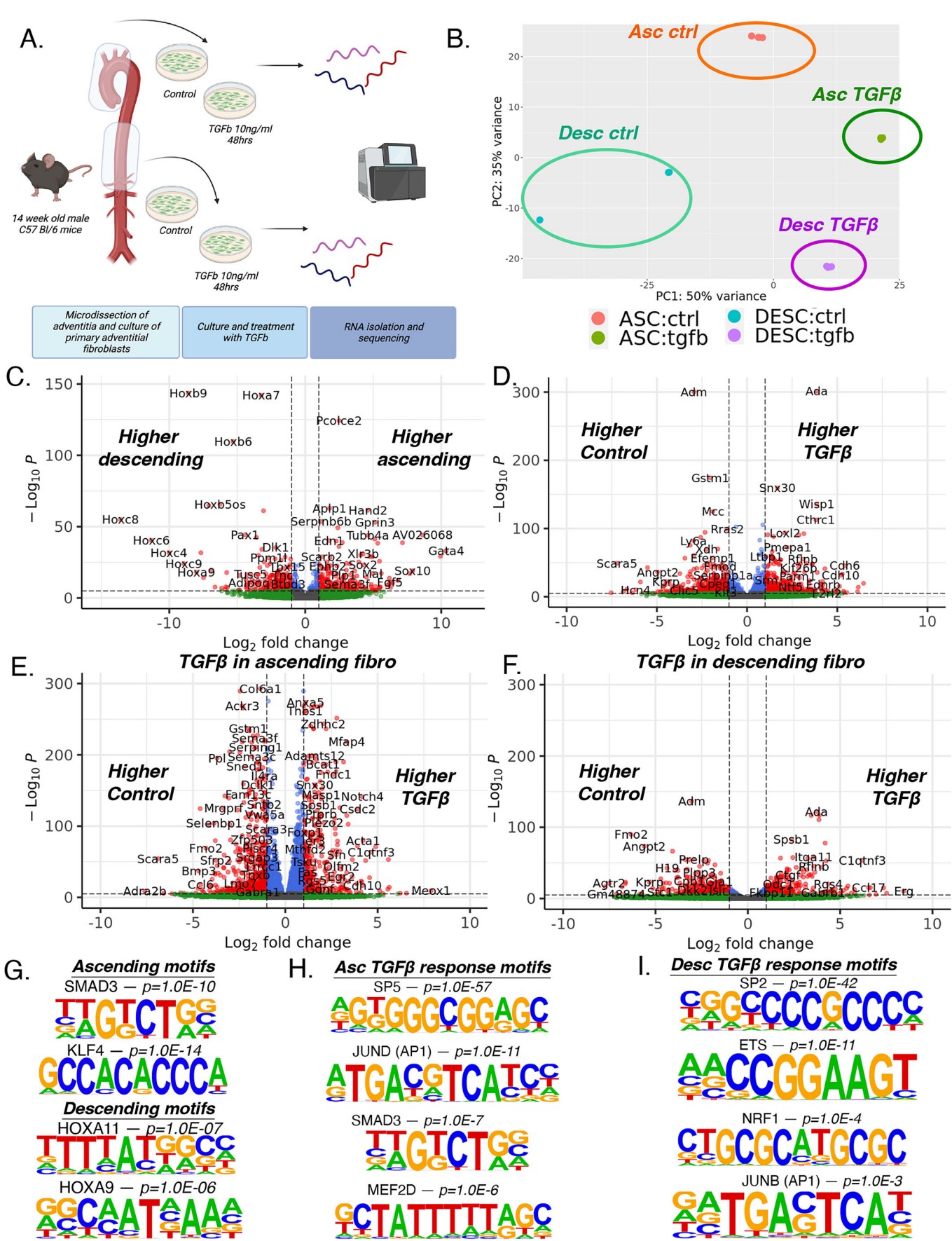

**Figure 6. Primary adventitial fibroblasts retain vascular site-specific transcriptomic features following in vitro culture revealing vascular site-specific response to TGFβ.**

Schematic diagram of primary adventitial fibroblast culture, treatment, and bulk RNA sequencing (**A**). Principal component analysis (PCA) by vascular site and TGFβ treatment (**B**). Volcano plots showing differential gene expression analysis of all samples by vascular site (**C**) and by TGFβ treatment (**D**). Volcano plot of DE gene analysis in response to TGFβ in ascending (**E**) and descending (**F**) fibroblasts. Motif sequences of identified enriched transcription factors of 200 bp sequences upstream of gene TSSs with increased expression in ascending and descending fibroblasts (**G**). Motif sequences of identified enriched transcription factors of 200 bp sequences upstream of gene TSSs in of upregulated genes in ascending fibroblasts in response to TGFβ treatment (**H**) and descending fibroblasts in response to TGFβ treatment (**I**). Differential gene expression was evaluated with DESeq2 using Wald test to assess the significance of $\log_2$ Fold Change and *P* values were adjusted for multiple testing using the Benjamini-Hochberg procedure. $N = 3$ control + ascending aorta, 3 TGFβ + ascending aorta, 3 control + descending aorta, 3 TGFβ + descending aorta.

To then understand how GWAS-associated gene sets may have enriched expression within cell types, we performed polygenic enrichment analysis using single-cell disease relevance score (scDRS) (Zhang et al, 2022). First, we generated the individual raw gene expression matrix from scRNAseq data including groups determined by cell type and vascular site. With summary statistics GWAS data from Pirruccello et al, (2022) we utilized the scDRS pipeline to evaluate GWAS gene enrichment for the ascending aortic dimension and descending aortic dimension GWASs. Following lifting the human gene sets to murine genome we then performed scDRS test of group-level statistics. We identified the associated Monte–Carlo *P* value (−log10) to represent the significance of cell type-disease association for ascending aortic dimension and descending aortic dimension (Fig. 7J,K). Here, we observed that for the ascending aortic dimension GWAS (Fig. 7J), the ascending SMC population had the strongest association, followed by carotid and descending aortic SMCs. Endothelial and fibroblast populations showed less strong of a relationship and macrophage cells had little to no association. The descending aortic dimension GWAS (Fig. 7K) similarly showed ascending SMC as the primary cell type of association, however overall P value associations were lower, likely influenced by a smaller number of genomic loci associated with descending aortic dimension. These data implicate the vascular SMC to be the primary mediator of complex genetic influence on aortic dimension trait.

## ChromBPnet predicts the human genotype effect of chromatin accessibility in a cell type and vascular site-specific manner

These data indicate that genes for which variants regulate aortic dimension have not only cell-type but also vascular site-specific expression and epigenomic patterns. This may suggest that a gene variant can have a differing effect on TF motif binding, chromatin accessibility, and gene regulation in not only a cell type but a vascular site-specific context. To better understand this vascular site-specific epigenomic regulation, we aimed to predict how a gene variant that influences the aorta may affect chromatin accessibility in a cell type and vascular site-specific manner using ChromBPNet (Pampari et al, 2025). ChromBPNet is a novel, bias-factorized, base-resolution deep learning model of chromatin accessibility (Pampari et al, 2025). Prior training of the ChromBPNet model has been predicated on the concept that gene variants affect TF binding of *cis*-regulatory elements (cREs) in a cell context-specific manner (Pampari et al, 2025). Training of a convolutional neural network (CNN) for ChromBPNet has utilized chromatin accessibility data across 5 ENCODE Tier 1 cell lines (Pampari et al, 2025), however, although this takes into account how a variant may affect TF

binding and chromatin accessibility within cell type, it does not allow for the investigation of a variant effect within cell type but across vascular sites. To overcome this barrier, we utilized our multiomic single-cell RNA and ATAC sequencing data and trained the ChromBPNet model in 9 different groups (Ascending aorta, Carotid, and Descending aorta, within SMC, Fibroblasts, and Endothelial cells) in our mouse dataset. Given this "genome-agnostic" approach, this model can predict the fundamental effect of basepair change on chromatin accessibility. These individual cell-type and vascular site-specific models can then be applied to the human genome. We leveraged the variants identified from the prior GWAS on aortic dimension (Pirruccello et al, 2022), performed linkage disequilibrium (LD) expansion to identify SNPs in LD with lead SNPs, and used each of the 9 models to score variant effect for 27,556 variants based on cell type and vascular site. We identified 469 high scoring aorta diameter LD-expanded GWAS variants across all of the vascular site datasets (example, rs2959350, in LD with lead SNP rs55736442 at *ANGPT1* locus, chr8:107,367,260:G:A, Ascending Endothelial cells, Fig. 8A). In this model, the variant allele has a pronounced activating effect on chromatin accessibility across a ~300 bp region.

For each variant, we scored the absolute log fold change (abs_logFC) from variant effect as well as the *P* value of the abs_logFC (abs_logFC.pval). To understand how predicted variant effect may vary within cell type across vascular site, we first evaluated the abs_logFC.pval and compared between ascending and descending SMC models (Fig. 8B). This reveals significant variation in *P* value of logFC for a given SNP between vascular sites (Fig. 8B). However, upon evaluation of the variation of abs_logFC for each SNP within SMC across vascular site, variation is significantly reduced (Fig. 8C). A similar level of variation of variant effect is seen when comparing ascending and descending fibroblast and endothelial cell models (Fig. 8D,E). To understand the predicted variant effect between cell types, we then compared the predicted abs_logFC in ascending SMC vs Fibro, SMC vs Endo, and Fibro vs Endo (Fig. 8F–H). We observe a consistent pattern of a greater degree of variation in comparing between cell types as opposed to within cell type across vascular site, consistent with an overall understanding that variant effect is primarily influenced by cell type, however, within cell type, vascular site continues to have a significant variant effect.

Following linear regression analysis of the abs_logFC within cell type between ascending and descending models (i.e., Asc SMC vs Desc SMC), we identified the top 1% of deviant SNPs (furthest away from linear regression line) based on quantile residuals. These top 1% deviant SNPs (a total of 274 SNP) highlight specific variants that appear to have vascular site-specific effect, and the 1% cutoff corresponds to a studentized *P* value residual value of 1.73E-4. For

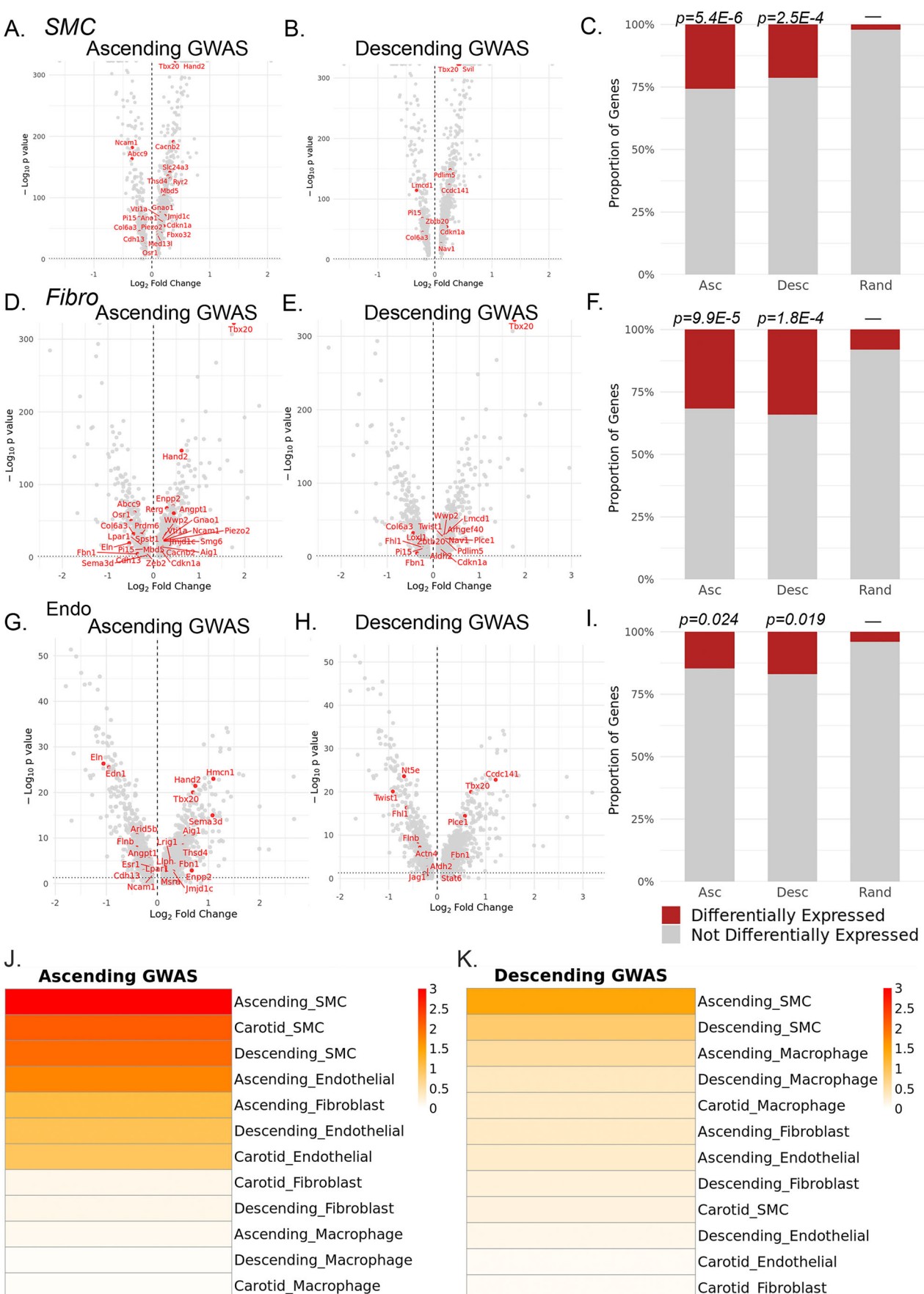

**Figure 7. Aortic dimension GWAS identifies vascular site-specific gene set enrichment.**

Volcano plots of ascending versus descending differential RNA expression and highlighting ascending and descending aortic dimension GWAS genes in SMC (**A, B**), fibroblast (**D, E**), and endothelial cells (**G, H**). Stacked barcharts showing the proportion of ascending/descending aortic dimension GWAS genes differentially expressed compared to a random gene list in SMCs (**C**), fibroblasts (**F**), and endothelial cells (**I**). (**J, K**) scDRS calculated Monte–Carlo −log10 *P* value for association to (**J**) ascending aortic dimension GWAS and (**K**) descending aortic dimension gene set enrichment analysis. Differential gene expression was evaluated with DESeq2 using Wald test to assess the significance of log$_2$ Fold Change and *P* values were adjusted for multiple testing using the Benjamini–Hochberg procedure. Enrichment of differentially expressed genes was conducted using chi-squared statistical test for significance compared to random 100 gene list selected from expressed genes within cell clusters. Differential gene expression testing represents 25,665 cells for SMC, 7586 cells for fibroblasts, and 1085 cells for endothelial cells.

example, rs11677932 is the lead SNP (chr2:237,315,312:G:A) at the *COL6A3* locus, a gene we have previously highlighted as having differential expression within cell type across vascular site. The variant effect is predicted to lead to epigenetic silencing; however, this effect is greatest in SMC, compared to fibroblast and endothelial cells, and within SMCs, this effect is greatest in descending SMC compared to ascending SMC (studentized outlier *P* value = 1.94E-33) (Fig. 8I). To then evaluate if our predicted silencing effect of this variant is reflected in larger human variant to gene expression databases, we evaluated the Genotype-Tissue expression (GTEx) database(Consortium, 2013) to determine if this variant (rs11677832) was an eQTL for *COL6A3*. We observed that rs11677832 is an eQTL for *COL6A3* (pval 6.75e-8) where the alternative allele A was associated with decreased *COL6A3* expression (Fig. 8J), consistent with our predictive modeling suggesting epigenetic silencing. To then define the proportion of loci predicted to have vascular site-specific effect, we identified 84 of 115 loci (73%) identified from Pirruccello et al, (2022) study, to have either lead SNPs or SNPs in LD with the lead variant to be predicted to have vascular site-specific effect in SMCs. Fourteen of which (12%) have 5 or more SNPs at the locus that meet our criteria for vascular site-specific effect. Top genes mapped to loci with vascular site-specific effect include *PRDM6*, *ULK4*, *ESR1*, *COL6A3*, *MSRA*, and *HAND2*.

The identification of variants that have vascular site-specific effects on chromatin accessibility may suggest that variants can change pioneer TF binding to motifs that influence vascular site-specific epigenetic patterns. To better understand these variants and the potential TF motifs that they regulate, we took the top 1% of deviant SNPs away from the linear regression, and from these SNPs, we selected a 200 bp genomic window (+/− 100 bp around each SNP) and performed unbiased Motif Discovery function with MEME Suite (Bailey et al, 2015). Known motif analysis of top-discovered motifs was then performed using TOMTOM (HOCO-MOCOv11) (Gupta et al, 2007). This analysis was performed for top deviant SNPs in SMC, fibroblast, and endothelial cell populations, and here, we identified novel motifs enriched in the genomic regions of these SNPs with differential effect on vascular site (Fig. 8K–N). TOMTOM analysis of discovered motifs revealed numerous TF motifs that we have similarly identified as having differential motif accessibility between vascular sites. For example, analysis of top-discovered motif from SMC data reveals motif enrichment for *MEF2A/C* (Fig. 8K), consistent with our prior finding of enriched *MEF2A/C* motif in descending SMC (Fig. 2N). Similarly, in analysis of the top-discovered motif from fibroblast data, we observe an enrichment for multiple zinc finger motifs including *IKZF1* and *ZN250*, however, we also see an enrichment for *TEAD4* (Fig. 8L), a motif we have previously observed to have

differential accessibility. Further, in the second top motif in this fibroblast data, we observe an enrichment for a *SMAD3* motif (Fig. 8M), consistent with our previously observed increase in *SMAD3* motif accessibility in ascending aortic fibroblasts. Finally, evaluation of these SNPs from endothelial data further reveals a discovered motif that is notably enriched for a *HAND2* motif (Fig. 8N), a motif we had similarly seen enriched in ascending aortic cell types. These data suggest that variants that have a differential effect on chromatin accessibility between vascular sites lie in genomic regions enriched for motifs that have differential accessibility between vascular sites. These data further support our hypothesis that vascular site-specific epigenomic patterns influence human genetic determinants of vascular disease risk.

# Discussion

Chromatin architecture and *cis*-regulatory elements such as enhancers and promoters are critical in mediating cellular gene programs in a cell type-specific manner (Anene-Nzelu et al, 2022). Disease-associated gene variants identified through GWAS are now increasingly being recognized to influence disease risk through modification of these regulatory regions of the genome (Maurano et al, 2012). In addition, there is now growing experimental evidence that vascular disease-associated gene loci influence disease risk in cell-type-specific mechanisms, where common human genetic variation can modify the function of cell-type-specific enhancers (Cheng et al, 2022; Wang et al, 2021). Risk of vascular diseases, such as atherosclerosis, aneurysm, or autoinflammatory vasculitides, are vascular site-specific, with disease-associated genetic loci suggesting differing genetic mechanisms of disease. These observations raise the question as to what extent cell-type-specific enhancers and gene programs vary by vascular site, and do these variations contribute to disease risk?

Here, we evaluated cell type and vascular site-specific enhancer and gene expression profiles of healthy vascular tissue in adult mice from three disease-relevant vascular sites, (1) ascending aorta and aortic root, (2) brachiocephalic and right common carotid arteries, and (3) descending thoracic aorta (Fig. 1A). These vascular sites represent the developmental diversity that makes up the aorta, with these regions arising from the secondary heart field and neural crest, neural crest, and somitic mesoderm, respectively. This work has revealed thousands of differentially accessible enhancers within vascular smooth muscle, fibroblasts, and endothelial cells. Through use of this single cell epigenomic and transcriptomic multiomic data, computational analysis of gene regulatory networks, in vitro culture of primary aortic fibroblast cells, and by using a novel machine learning approach to train and predict gene variant effect

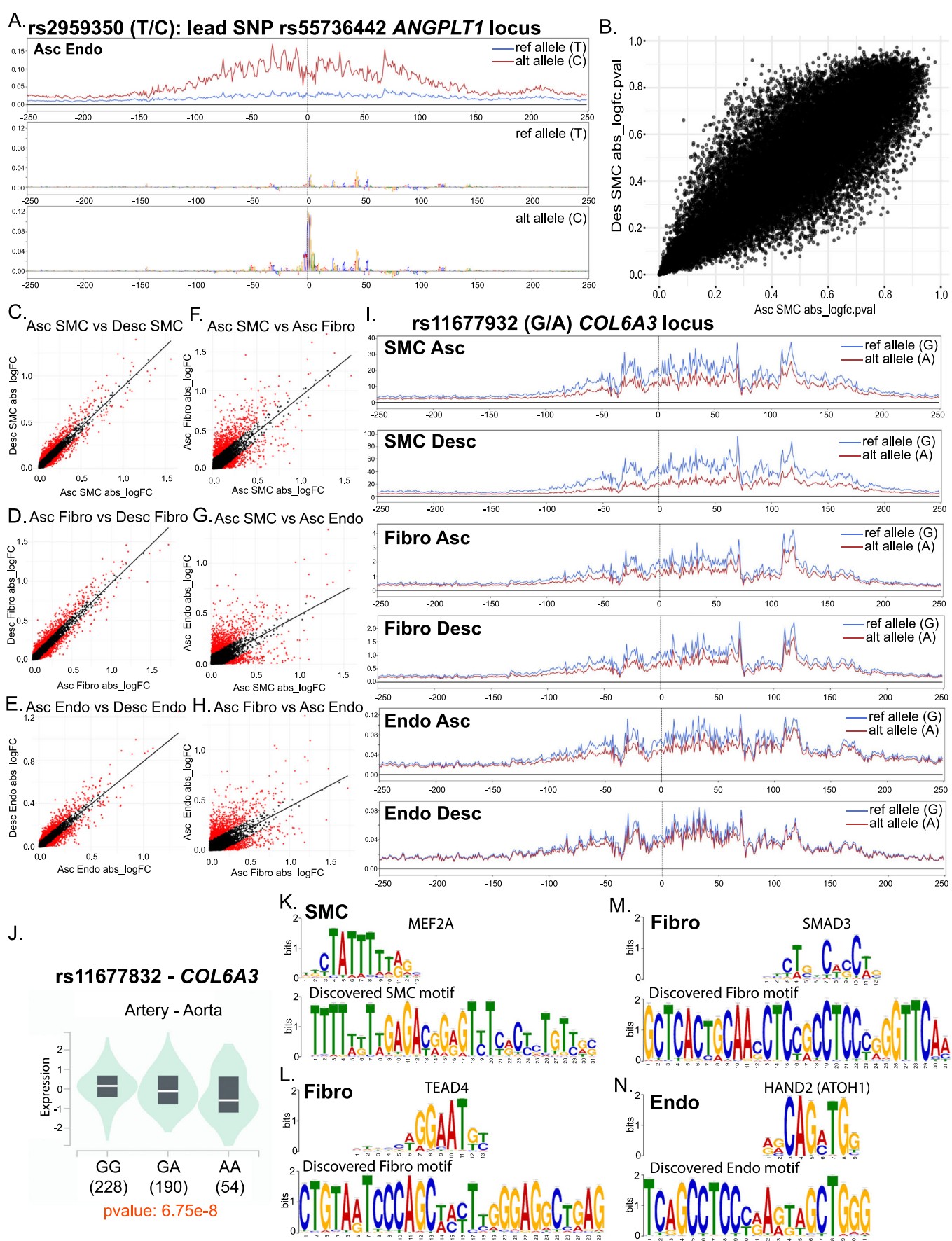

**Figure 8. ChromBPNet machine learning models predict variant effect on chromatin accessibility to be cell type and vascular site-specific.**

Example locus of ChromBPNet prediction of chromatin accessibility in reference and alternative alleles for rs2959350 at *ANGPLT3* locus with predicted basepair contributing effects (**A**). Scatter plot of absolute log fold change (abs_logFC) *P* values for scored variants in ascending SMC versus descending SMC models (**B**). Scatter plots of abs_logFC for scored variants in ascending SMC versus descending SMC models (**C**), ascending fibro versus descending fibro models (**D**), ascending endothelial versus descending endothelial models (**E**), ascending SMC versus ascending fibroblast models (**F**), ascending SMC versus ascending endothelial models (**G**), and ascending fibroblast versus ascending endothelial models (**H**). Example locus of ChromBPNet prediction of chromatin accessibility in reference and alternative alleles for rs11677932 at *COL6A3* locus for ascending and descending SMCs, fibroblasts, and endothelial cell models (**I**). GTEx eQTL expression data for rs11677932 from human aortic arterial tissue for *COL6A3* (**J**). Examples of discovered and matched motifs in SMC—*MEF2A* (**K**), Fibroblasts—*TEAD4* (**L**), Fibroblasts—*SMAD3* (**M**), and Endothelial —*HAND2* (**N**). Boxplot in (**J**) represents the interquartile range (median with 25th and 75th percentile) with lower and upper whisker representing the lowest and highest data point within 1.5× IQR below or above the Q1 and Q3.

on chromatin accessibility across vascular sites in the human genome (ChromBPNet), we have defined an important observation —that epigenomic patterns are not only cell type but vascular site specific. This data supports our understanding that gene variants appear to regulate chromatin accessibility through influencing pioneer TF motif binding of key TFs (i.e., *SMAD3*, *MEF2A/C*, *HAND2*, *TEAD4*) in a vascular site-specific mechanism. These data have important implications for our understanding of vascular site-specific disease risk and may give insight into novel mechanisms of disease.

The activation of key developmental transcription factors is crucial in the coordinated cellular development of the fetal and adult vasculature. Although expression of genes such as *Hand2*, *Tbx20*, *Gata4*, *Wnt* and those encoding related WNT signaling molecules, and *Hoxa/b/c* family of transcription factors have previously been known to mediate vascular development (Anderson et al, 2008; Aquino et al, 2021; Li et al, 2012; Majesky, 2007; Paffett-Lugassy et al, 2013; Sawada et al, 2017), the residual chromatin accessibility and gene expression of these developmental transcription factors in the adult vasculature has been less well characterized. A notable observation of our study is that key regulatory enhancers of development genes *Tbx20*, *Hand2*, and *Gata4* have increased chromatin accessibility in the ascending aorta and carotid artery in comparison to the descending aorta in smooth muscle and fibroblasts (Figs. 2 and 3). Whereas *Hoxb* and *Hoxa/Hoxc* family of transcription factors have increased chromatin accessibility in the descending aorta (Figs. 2 and 3). Although it is possible that hemodynamic factors may be contributing to these epigenetic differences across vascular sites, our data from culturing primary adventitial fibroblasts in vitro suggests that these transcriptomic patterns are retained even when culturing in a dish and across three passages (Fig. 6). These findings suggest that these vascular cells retain an epigenetic "memory" of their developmental program, suggesting these cells are poised to turn on these gene programs.

This finding further raises the question as to the role of these development genes in mediating vascular disease risk in adulthood. Although genetic variation in *TBX20* has been associated with a spectrum of congenital cardiac lesions relating to cardiac and vascular development (Chen et al, 2021), there have been increasing observations that genetic loci near *TBX20* meet genome-wide significance for disease in adulthood including coronary artery disease (Koyama et al, 2020), myocardial infarction (Sakaue et al, 2021), blood pressure (Warren et al, 2017), and aortic dimension and distensibility (Benjamins et al, 2022; Pirruccello et al, 2022). Similarly, variants near *GATA4* and members of the *HOXB* family of transcription factors such as *HOXB7* have been associated with

hypertension (Kichaev et al, 2019; Takeuchi et al, 2018), while *HAND2* variants are associated with aortic dimension and atrial fibrillation (Pirruccello et al, 2022; Roselli et al, 2018). The vascular site and cell type-specific chromatin accessibility observed in our study of these developmental transcription factors may suggest their ongoing role in mediating vascular disease in adulthood in a vascular site-specific mechanism. This hypothesis of developmental TFs involved in vascular disease pathogenesis has been similarly supported by work from our lab on developmental TFs *TCF21* and *ZEB2* (Cheng et al, 2022; Wirka et al, 2019). These TFs—which have important roles in vascular development, regulating cell state transitions and endothelial to mesenchymal transition (Acharya et al, 2012; Fardi et al, 2019)—have been identified through GWAS as having additional roles in the development of CAD (Erdmann et al, 2018; Nikpay et al, 2015). SMC-specific deletion of *Tcf21* and *Zeb2* in the mouse revealed a significant effect on transcriptional regulation, epigenetic landscape, and plaque characteristic, further defining their roles as causal CAD genes (Cheng et al, 2022; Wirka et al, 2019).

TF motif accessibility analysis identified an important observation that *SMAD2:SMAD3* motif is enriched in ascending aortic fibroblasts, highlighting a potential vascular site-specific response to TGFβ signaling. It was first identified nearly 30 years ago in chick embryos that vascular SMCs derived from the neural crest have enhanced response to TGFβ signaling compared to ectoderm-derived SMCs (Gadson et al, 1997; Topouzis and Majesky, 1996). More recently, transcriptomic differences across vascular sites and organ-specific fibroblast transcriptomic identity have been reported (Forte et al, 2022; Yu et al, 2022). However, epigenomic landscape and differential motif accessibility across vascular sites has not been defined. By isolating and culturing primary adventitial fibroblasts from ascending and descending aorta, in vitro experiments with TGFβ stimulation with bulk RNA sequencing confirmed that differential gene expression by vascular site is retained following isolation and in vitro culture and revealed ascending fibroblasts to have a markedly greater response to TGFβ than descending fibroblasts (Fig. 6). Motif accessibility analysis of differentially expressed genes further suggests distinct response to AP1 factors such as JUN, SMAD2/3, and MEF2 factors (Fig. 6H). These are two important observations that reveals, (1) DE genes by vascular site in fibroblasts are independent of differential laminar and turbulent flow and (2) implicates differential chromatin accessibility and relevant TF motif accessibility to have causal biological and disease-relevant function.

Evaluation of GRNs on a cell type and vascular site-specific basis highlights the ascending aortic fibroblasts to have a notable activation of *Meox1* as a key GRN TF (Fig. 5). *MEOX1* is a master

regulator of fibroblast activation *Meox1* (Alexanian et al, 2021) and we further identified an epigenetic "priming" of key disease-relevant genes. Primary fibroblast culture shows similar baseline RNA expression of *Meox1* between vascular sites, but *Meox1* is upregulated to a much greater extent in ascending fibroblasts in response to TGFβ (Appendix Fig. S6C), consistent with epigenetic priming and activation of key vascular site-specific GRN.

Prior work has discovered that genomic variants identified through GWAS have cell-type-specific effects, however, understanding if gene variants may have not only a cell type but vascular site-specific effect is challenging to determine. Our data here highlights that many of the GWAS genes identified from a recent aortic dimension GWAS (Pirruccello et al, 2022) are differentially expressed when comparing ascending versus descending aortic SMC, fibroblasts, and endothelial cells, with a notable enrichment of these genes in the fibroblast DE gene analysis (Fig. 7). An additional recent GWAS on aortic dissection (DePaolo et al, 2025) further highlights loci that appear to have differential expression across vascular sites (i.e., *ARHGAP31* that we observe to have differential expression in fibroblasts across vascular sites). However, to understand how these variants may regulate chromatin accessibility on a basepair resolution, we applied a unique model, whereby training a ChromBPNet model in each cell type and vascular site, we employed 9 distinct models to predict variant effect of aortic dimension on a cell type and vascular site-specific context. This work revealed that although cell type remains the primary influence on chromatin accessibility, the vascular site is an important influence. Our work further highlights that variants that appear to modify effect based on vascular site are enriched for genomic positions that have key TF motifs, suggesting that these variants may modify the binding of TFs such as *SMAD3*, *MEF2A*, and others and regulate disease risk in a vascular site-specific mechanism (Fig. 8).

## Limitations

In this study, our aim was to evaluate single-cell enhancer and transcriptional profiles across vascular sites in healthy tissue. A limitation of this study is that we did not evaluate the epigenomic profiles in a disease state across vascular sites. However, by isolating and culturing primary adventitial fibroblasts and evaluating differential response to TGFβ, we gained further insight into disease-relevant mechanisms. We anticipate that there are dynamic changes in chromatin accessibility and gene expression programs in disease, as has been previously observed (Cheng et al, 2022; Wang et al, 2021), and future studies will be aimed at understanding how these dynamic chromatin accessibility changes occur across vascular sites. Similarly, recent work has demonstrated changes in chromatin accessibility in vascular tissue with aging (Xie et al, 2022). Our study selected for adult mice that are 14–16 weeks of age, which represents a young adult. We anticipate that chromatin accessibility dynamically changes with aging, and future studies to evaluate how this change differs across vascular sites may be particularly insightful.

## Conclusions

By performing combined scRNAseq and scATACseq on vascular tissue across three vascular sites, we reveal for the first time that the epigenomic landscape and transcriptional profiles of vascular smooth muscle, fibroblasts, and endothelial cells, are specific to anatomic origin, that genomic regions and genes that differentiate cells by vascular site are weighted towards developmental genes, and that vascular cells have an epigenetic "memory" of their developmental program. We discovered that differential chromatin accessibility appears to "prime" vascular site-specific gene regulatory networks in disease-relevant mechanisms, and finally, that genetic variants that influence aortic dimension appear to regulate chromatin accessibility in not only a cell type, but also a vascular site-specific mechanism. This work supports the paradigm that genetic mechanisms of disease influence disease risk in a vascular site-specific manner, gives unique insight into vascular site-specific transcriptional and epigenetic programs, and further creates a valuable single-cell atlas for the vascular biology community.

## Methods

### Reagents and tools table

| Reagent/resource | Reference or source | Identifier or catalog number |
|---|---|---|
| **Experimental models** | | |
| C57BL/6J (*M. musculus*) | Jackson Labs | Strain #000664 |
| **Recombinant DNA** | | |
| NA | | |
| **Antibodies** | | |
| NA | | |
| **Oligonucleotides and other sequence-based reagents** | | |
| 10X Genomics, Single cell RNAseq reagents for single cell capture and library preparation. 3′ capture V3.1 chemistry reagents. | 10X genomics | https://www.10xgenomics.com/support/universal-three-prime-gene-expression/documentation/steps/library-prep/chromium-single-cell-3-reagent-kits-user-guide-v-3-1-chemistry |
| 10X Genomics, Single cell ATACseq reagents for single nuclear capture and library preparation. V2 reagents. | 10X genomics | https://www.10xgenomics.com/support/epi-atac/documentation/steps/library-prep/chromium-next-gem-single-cell-atac-v-2-reagents-workflow-and-data-overview |
| **Chemicals, enzymes, and other reagents** | | |
| **Software** | | |
| R version 4.2.2 | | https://www.r-project.org/ |
| Prism 10 | | https://www.graphpad.com/ |
| **Other** | | |

## Mice and microdissections

Male 14–16-week-old C57Bl/6 mice were purchased from Jackson Laboratory (Bar Harbor, ME). The animal study protocol was approved by the Administrative Panel on Laboratory Animal Care

at Stanford University, and procedures were followed in accordance with institutional guidelines.

Mice were anesthetized with isoflurane and sacrificed with the cervical dislocation technique. Vascular tissue was flushed with injection of 5 mL of phosphate-buffered saline (PBS) into the left ventricle after an incision was made at the right atrium. Aortic tissue was dissected including the aortic root and ascending aorta up to the take-off of the brachiocephalic artery. The brachiocephalic artery and its extension into the right common carotid artery were carefully dissected under a stereoscope. The descending thoracic aorta was then isolated from past the left subclavian artery down to the renal arteries.

## Vascular tissue dissociation, cell capture, and sequencing

Tissues were collected and dissociated for single cell capture as previously described (Cheng et al, 2022; Kim et al, 2020; Wirka et al, 2019). Briefly, vascular tissue was washed three times in PBS, tissues were then placed into an enzymatic dissociation cocktail (2 U ml$^{-1}$ Liberase (5401127001; Sigma–Aldrich) and 2 U ml$^{-1}$ elastase (LS002279; Worthington) in Hank's Balanced Salt Solution (HBSS)) for 45 min at 37 °C. Tissues were then gently minced and dissociated with a pipette. The cell suspension was strained and then pelleted by centrifugation at 500 × $g$ for 5 min. The enzyme solution was then discarded, and cells were resuspended in fresh HBSS. To increase biological replication, two sets of 8 mice (16 mice total) were used to obtain two single-cell suspensions for each vascular tissue (aortic root/ascending aorta, brachiocephalic/carotid, and descending thoracic aorta). Cells were FACS sorted, and live cells were identified as previously described (Wirka et al, 2019). Cells were sorted on a Sony SH800S cell sorter, where cells were gated on forward/side scatter parameters to exclude small debris and then gated on forward scatter height versus forward scatter area to exclude obvious doublet events. Approximately 100–150,000 live cells were sorted for each vascular site for each capture, where a portion of cells were taken directly to scRNAseq capture. For single-cell ATAC, cells were collected in BSA-coated tubes, and nuclei isolated per 10X recommended protocol, and captured on the 10X scATAC platform.

All single-cell captures, and library preparation was performed in the Quertermous Lab. Cells were loaded into a 10x Genomics microfluidics chip and encapsulated with barcoded oligo-dT-containing gel beads using the 10x Genomics Chromium controller according to the manufacturer's instructions. Single-cell libraries were then constructed according to the manufacturer's instructions (Illumina). Libraries from individual samples were multiplexed into one lane before sequencing on an Illumina platform with targeted depth of 50,000 reads per cell for RNA and 75,000 reads/cell for ATAC. Sequencing was performed by MedGenome (Foster City, CA). Post filtering for non-cells, mean number of reads within peaks per cell in scATAC data was 18,000–20,000 as was seen in our prior report (Cheng et al, 2022).

## Analysis of single-cell sequencing data

### scRNAseq

Fastq files from each vascular site (6 total RNA captures) were aligned to the reference genome (mm10) individually using CellRanger Software (10x Genomics). Dataset was then analyzed and captures were integrated using the R package Seurat (Stuart et al, 2019). The dataset was trimmed of cells expressing fewer than 1000 genes, and genes expressed in fewer than 50 cells. The number of genes, number of unique molecular identifiers and percentage of mitochondrial genes were examined to identify outliers. As an unusually high number of genes can result from a "doublet" event, in which two different cell types are captured together with the same barcoded bead, cells with >7500 genes were discarded. Cells containing >7.5% mitochondrial genes were presumed to be of poor quality and were also discarded. QC of nFeature_RNA, nCount_RNA, and percent.mt are included in Appendix Fig. S7A–C. The gene expression values then underwent library-size normalization and normalized using established Single-Cell Transform function in Seurat. Principal component analysis was used for dimensionality reduction, followed by clustering in principal component analysis space using a graph-based clustering approach via Louvain algorithm. Batch correction was performed with reciprocal PCA (RPCA). UMAP was then used for two-dimensional visualization of the resulting clusters. Analysis, visualization and quantification of gene expression and generation of gene module scores were performed using Seurat's built-in function such as "FeaturePlot", "VlnPlot", and "FindMarker."

For GWAS gene set enrichment analysis on single cell resolution, we performed single-cell disease relevance score (scDRS) as previously described (Zhang et al, 2022). Briefly, we generated a raw expression matrix from scRNAseq data based on the 12 groups of cell type (SMC, Fibro, Endo, Macs) and vascular site (Ascending, Carotid, Descending) (i.e., Ascending_SMC, Carotid_SMC, etc.). Utilizing summary statistic GWAS data for aortic dimension (ascending and descending aortic dimension GWAS (Pirruccello et al, 2022)) and lifting human genome genes to murine genome, we performed scDRS analysis of disease enrichment (aortic dimension trait) for individual cells followed by scDRS test for group-level statistics. Monte–Carlo $P$ value of group-level association to disease trait was reported for significance.

### scATACseq

Fastq files from each vascular site (six total ATAC captures) were aligned to the reference ATAC genome (mm10) individually using CellRanger Software (10x Genomics). Individual datasets were aggregated, and peak calling was performed using the CellRanger aggr command without subsampling normalization. The aggregated dataset was then analyzed using the R package Signac (Stuart et al, 2019). The dataset was first trimmed of cells containing fewer than 1000 peaks, and peaks found in fewer than 10 cells. The subsequent cells were then again filtered based off TSS enrichment, nucleosome signal, and percent of reads that lies within peaks found within the larger dataset. Cells with greater than 20,000 reads within peaks or fewer than 3000 peaks, <2 transcription start site (TSS) enrichment, or <15% reads within peaks were removed as they are likely poor-quality nuclei. QC of pct_reads_in_peaks, peak_region_fragments, TSS.enrichment, and nucleosome_signal are included in Appendix Fig. S7D–G. Fragment histograms based on high and low nucleosome group (NS > 4, NS < 4) and TSS Plot based on high and low TSS enrichment (TSS > 2, TSS < 2) are included in Appendix Fig. S7H,I. The remaining cells were then processed using RunTFIDF(), RunSVD() functions from Signac to allow for latent semantic indexing (LSI) of the peaks (Cusanovich

et al, 2015; Cusanovich et al, 2018), which was then used to create UMAPs. Batch correction for UMAP visualization was performed with Harmony. Differentially accessible peaks between different populations of cells were found using FindMarker function, using number of peaks as latent variable to correct for depth. Motif matrix was obtained from JASPAR 2020, aligned onto BSgenome.Mmusculus.UCSC.mm10. Accessibility analysis around transcription factor motifs was performed using ChromVar (Schep et al, 2017). Merging of scRNA and ATAC data was performed using the Pseudo-expression of each gene created using scATACseq GeneActivity function to assign peaks to nearest genes expressed in the scRNA dataset, and mapped onto each other using Canonical Correlation Analysis. For gene regulatory network (GRN) analysis, RNA and ATAC integrated datasets were subsampled based on cell type and GRNs were inferred using Pando (Fleck et al, 2023). In this method, variable features are identified from RNA dataset and GRN is initiated with 'initiate_grn()' function. Candidate regions for TF-binding motifs are identified and GRNs are inferred using 'infer_grn()' function. GRN modules are then identified using 'find_modules()' function for each cell type and vascular site reported. Network graphs are then visualized using 'plot_network_graph()' function and further analysis was performed following standard Pando workflow.

## ChromBPNet

### Processing of single-cell ATAC-seq data

Methods for ChromBPNet model has been recently described in detail (Marderstein et al, 2025; Pampari et al, 2025). For each cell type and vascular site cluster (i.e., SMC/Fibro/Endo, Asc/Car/Desc) from the dataset, we created pooled all of the fragments that belonged to each cluster based on barcode to cell type maps to create pseudobulk fragment files. We then splitting each fragment in half to obtain one read per strand and converted each fragment file to a tagalign file. We then used these files as input to the ENCODE ATAC-seq pipeline v1.10.0 (available at https://github.com/ENCODE-DCC/atac-seq-pipeline), which generated peak calls using MACS2 (available at https://github.com/macs3-project/MACS). For downstream analyses involving ChromBPNet, we used the overlap peak set, which only includes peaks called from the original sample that overlap with peak calls from both of the pseudo-replicates created by the pipeline by randomly allocating the reads from the original sample into each synthetic replicate.

### Training ChromBPNet models

To model the cell type and vascular-specific chromatin accessibility in each vascular bed cell type, we trained ChromBPNet models on their cell type-resolved pseudobulk ATAC-seq profiles (ChromBPNet is available at https://github.com/kundajelab/ChromBPNet). For each sample, we used as input the peak and tagalign files generated by the ENCODE ATAC-seq pipeline, along with a Tn5-bias model trained on the descending aorta SMC. We trained the ChromBPNet models using a fivefold cross-validation scheme—ensuring that each chromosome appeared in the test set of at least one cross-validation fold. Dataset EV25 outlines the chromosomes included in the train, validation, and test splits for each fold in the mouse.

Upon completion of training, any model that continued to respond to the tn5 motif's consensus sequences after unplugging the bias model, indicating an unsuccessful bias-correction procedure, was re-trained until the model showed a limited response to the tn5 motif's consensus sequences embedded in random genomic backgrounds. Specifically, this retraining was performed if the maximum prediction from the profile head for any basepair from the examples containing the tn5 motif exceeded 0.002.

### Creating the variant list

To create a comprehensive set of variants associated with aorta diameter, we first obtained all of the genome-wide significant GWAS variants from Pirruccello et al ($P$ value < 5e-8). Next, we used Plink v1.9 to find all variants from the AFR, AMR, EAS, EUR, and SAS populations in 1000 genomes that are in high LD ($r^2 > 0.8$) with the genome-wide significant GWAS variants, and we used TopLD to do the same for the African, East Asian, European, and South Asian populations in TOPMed. The final variant list included all of the genome-wide significant variants and any variants in high LD with them from any of the above populations in either 1000 Genomes or TOPMed.

## Predicting variant effects using ChromBPNet models

To score each variant in this study, we used the ChromBPNet models for each cell type to predict the base-resolution scATAC-seq coverage profiles for the 1 kb genomic sequence centered at each variant and containing the reference and alternate allele. We then estimated the variant's effect size using two measures: (1) the log2 fold change in total predicted coverage (total counts) within each 1 kb window for the alternate versus reference allele and (2) the Jensen–Shannon distance (JSD) between the base-resolution predicted probability profiles for the reference and alternate allele (capturing changes in profile shape).

We assessed statistical significance for these scores using empirical null distributions constructed by shuffling the 2114 bp sequence around each variant multiple times while preserving dinucleotide frequency. Next, each shuffled sequence was duplicated, and the variant's reference or alternate allele was inserted at the center, resulting in a total of one million null variants for each set of observed variants scored. Each null variant was scored with the same procedure as the observed variants. For each observed variant, we then computed the proportion of null variants with an equally high or higher (more extreme) score to derive empirical $P$ values for the log2 fold change and JSD scores. The code base for scoring variants is at https://github.com/kundajelab/variant-scorer.

### Motif enrichment analysis from scored ChromBPNet variants

To identify variants that have a differential effect based on vascular site (i.e., between ascending and descending aortic cell types), we performed linear regression analysis for absolute log fold change between all scored variants between ascending and descending SMC, fibroblasts, and endothelial cell ChromBPNet models. We identified the top 1% of variants that "deviate" from the linear regression line that represent the top 1% of variants that have a differing effect based on vascular site within cell type. The genomic position of this top 1% variant list was expanded 100 bp upstream and downstream to have a 200 bp window that was converted to a BED file. These genomic positions were then used unbiased Motif Discovery function with MEME Suite (Bailey et al, 2015). Known

motif analysis of top-discovered motifs was then performed using TOMTOM (HOCOMOCOv11) (Gupta et al, 2007).

## Primary adventitial fibroblast culture and bulk RNA sequencing

We isolated the aortic root and ascending aorta from 6, 12-week-old male C57/BL6 mice as well as the descending thoracic aorta to the level of the renal arteries and carefully separated the media from the adventitia by gentle traction with forceps. Adventitia was cut into small pieces with microscissors and plated into 12-well plates with DMEM media with 5% FBS. Following cell attachment to the plate, cells were cultured at 37 C and expanded for 48 h. Cells were washed with PBS and passaged 1× and plated into a 12-well plate at 65,000 cells/mL. Upon cell culture reaching ~80% confluence, cells were treated with either solute control or mouse TGFβ (invitae) (10 ng/mL) for 48 h. Following 48 h treatment, RNA was isolated using Qiagen RNeasy kits and RNA/cDNA library was sequenced with 250 M paired end reads. Fastq files were aligned to MM10 murine genome and raw read count and expression normalization was performed with featureCounts (Liao et al, 2014) and DEseq2 (Love et al, 2014), respectively. Using DEseq2 standard pipeline, we performed differential gene expression analysis between groups (i.e., Asc-control versus Desc-control; Asc-TGFβ versus Desc-TGFβ; Asc-control versus Asc-TGFβ; Desc-control versus Desc-TGFβ). Further interaction analysis was performed where DESeq2 pipeline evaluates genes whose response to TGFβ is not the same in both regions—genes that are regulated by TGFβ in one aortic region but not the other, or that change in opposite directions, helping identify region-specific responses to TGFβ. Differential predicted transcription factor motif accessibility from bulk RNAseq analysis of the 200 bp upstream of the transcription start site (TSS) from differentially expressed genes was performed with Hypergeometric Optimization of Motif EnRichment (HOMER) (Heinz et al, 2010).

## Data availability

All scRNAseq, scATACseq, and bulk RNAseq data have been deposited to the National Center for Biotechnology Information Gene Expression Omnibus under GEO accession numbers GSE296197 (scRNA/ATACseq) and GSE296074 (bulk RNAseq).

The source data of this paper are collected in the following database record: biostudies:S-SCDT-10_1038-S44320-025-00140-2.

## Peer review information

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

## Acknowledgements

This work was supported National Institutes of Health grants F32HL160067 (CW), L30HL159413 (CW), K08HL167699 (CW), K08HL153798 (PC), R01HL179083 (PC), R01HL134817 (TQ), R01HL139478 (TQ), R01HL156846 (TQ), R01HL151535 (TQ), R01HL145708 (TQ), UM1HG011972 (TQ), and K08HL177251 (BTP). This work was also supported by American Heart Association grants 20CDA35310303 (PC), 23CDA1042900 (CW), 23POST1018991 (WG), 24POST1187860 (JPM), 24SCEFIA1248386 (PC), and 24CDA1272805 (BTP). We also thank the Chan Zuckerberg Foundation, Human Cell Atlas Grant (ZF2019-002437) (TQ and PC).

## Author contributions

**Chad S Weldy**: Conceptualization; Data curation; Software; Formal analysis; Funding acquisition; Validation; Investigation; Visualization; Methodology; Writing—original draft; Project administration; Writing—review and editing. **Soumya Kundu**: Software; Formal analysis; Investigation; Methodology. **João Monteiro**: Data curation; Methodology. **Wenduo Gu**: Investigation; Methodology. **Albert J Pedroza**: Investigation; Methodology. **Alex R Dalal**: Investigation; Methodology. **Matthew D Worssam**: Investigation; Methodology. **Daniel Li**: Software; Investigation; Methodology. **Brian Palmisano**: Investigation; Methodology. **Quanyi Zhao**: Investigation; Methodology. **Disha Sharma**: Investigation; Methodology. **Trieu Nguyen**: Investigation; Methodology. **Ramendra Kundu**: Investigation; Methodology. **Michael P Fischbein**: Funding acquisition; Project administration. **Jesse Engreitz**: Supervision. **Anshul B Kundaje**: Supervision. **Paul P Cheng**: Supervision; Funding acquisition; Investigation; Methodology. **Thomas Quertermous**: Supervision; Funding acquisition; Methodology.

Source data underlying figure panels in this paper may have individual authorship assigned. Where available, figure panel/source data authorship is listed in the following database record: biostudies:S-SCDT-10_1038-S44320-025-00140-2.

## Disclosure and competing interests statement

TQ is a scientific advisor for Amgen. CSW is a consultant and advisory board member for Avidity Biosciences and AIRNA Bio. The remaining authors declare no competing interests.

