## [Peer Review File · Molecular Systems Biology]

Epigenomic landscape of single vascular cells reflects developmental origin and disease risk loci

Chad Weldy, Soumya Kundu, João Monteiro, Wenduo Gu, Albert Pedroza, Alex Dalal, Matthew Worssam, Daniel Li, Brian Palmisano, Quanyi Zhao, Disha Sharma, Trieu Nguyen, Ramendra Kundu, Michael Fischbein, Jesse Engreitz, Anshul Kundaje, Paul Cheng, and Thomas Quertermous

Corresponding author(s): Chad Weldy (weldyc@stanford.edu) , Thomas Quertermous (tomq1@stanford.edu)

Review Timeline:

Submission Date:	24th Jul 25
Editorial Decision:	13th Aug 25
Revision Received:	14th Aug 25
Accepted:	19th Aug 25

Editor: Jingyi Hou

Transaction Report:

Please note that the manuscript was transferred from another journal where it was originally reviewed.

Response to reviewer comments for Weldy et al., (MSB-2025-13237-T)

Dear Dr. Hou,

We are thrilled to have our manuscript now under consideration at EMBO Molecular Systems Biology and we appreciate your interest in our work. As detailed below, we have made substantial modifications to our manuscript that includes 1) additional analysis of GWAS gene set enrichment analysis using scDRS including new figure panels to Figure 7; 2) new eQTL analysis of prioritized SNP variants and new figure panels to Figure 8; 3) new GRN figure and clarification within the ascending fibroblast population with a new supplemental figure 4; 4) new statistical analysis of chromBPnet prioritized variants to provide additional clarity on statistical significance; as well as 5) new text and citations to provide additional clarity to the reader.

We feel that this manuscript has improved significantly, and we appreciate the reviewers thoughtful and timely feedback for this manuscript.

Reviewer #1 (Remarks to the Author):

The authors have made commendable progress in addressing prior concerns, particularly by incorporating additional replicates and improving the structure and clarity of the text. While the manuscript presents extensive and well-integrated data on chromatin accessibility, gene expression, and inferred regulatory networks, the functional roles of several key genes remain largely speculative and would benefit from further validation.

• While the authors present rich descriptive data, the functional roles and clinical or biological relevance of key genes remain insufficiently explored. A minimal experimental validation, such as CRISPRi/a or siRNA perturbation of one or two key transcription factors (e.g., Meox1, Hand2) in site-specific fibroblasts, could substantially strengthen the manuscript. This would confirm whether vascular-site specific epigenetic priming is functionally meaningful in shaping gene expression programs or cellular responses (e.g., to TGF β).

We thank the reviewer for their kind words regarding our efforts to address prior concerns. We indeed have put significant effort toward expanding the number of replicates, we worked to clarify the text, and added a significant number of new computational approaches. We do appreciate and understand the comment regarding how additional experimental validation would provide strength to the manuscript. While performing additional experiments could possibly provide some new and helpful insights, there are two primary concerns that have tempered our enthusiasm for proceeding with these experiments.

First, our data implicates multiple differential gene regulatory networks influenced by epigenomic programs that mediate vascular site-specific biological responses. In the primary fibroblast experiment, we presented our data showing increased susceptibility to TGFb in ascending fibroblasts compared to descending fibroblasts. Our data may implicate multiple factors that influence this, including differential SMAD3 motif accessibility, MEOX1, key TFs such as KLFs and others. One concern is that performing targeted gene KD (either CRISPRi or siRNA) of a single gene, particularly if it is a key TGFb responsive TF such as SMAD3, may show effect on response to TGFb, but interpreting that finding at a single gene approach will be limited. For example, KD of SMAD3 would be expected to inhibit canonical TGFb response in any fibroblast, and to interpret a blunted TGFb response as experimental validation to support our finding may be limited. This type of experiment will inherently be limited and falls outside the efforts of this current manuscript to describe broad changes in regulatory networks that imply multiple genes outside of single drivers of biological difference.

Second, at a technical standpoint, the method to isolate primary adventitial fibroblasts from the mouse aortic segment, particularly from the ascending aorta, and culture in enough quantity to have adequate samples for experimental treatment with replication is technically challenging. Our group had multiple efforts involving nearly 20 mice to get to where we could get the replicates that we did have for the TGFb experiment we included in the manuscript. To perform these additional validation experiments we would need to plate primary advFibros from ascending and descending aorta in sufficient number to be able to perform siRNA control, siRNA target gene(s), siRNA control + TGFb, and siRNA target gene(s) + TGFb. This has inherent technical challenges and is beyond the scope of the current work.

• The addition of ChromBPNet modeling to evaluate variant effects in a vascular bed-specific context is a useful and welcome step forward. However, while the model predicts how variants may impact chromatin accessibility, it does not directly inform on gene expression changes, which limits the downstream biological interpretation. This could be a valuable opportunity to further strengthen the functional characterization, for example by integrating datasets such as eQTLs for one or two well-supported loci, if feasible.

We agree with the reviewer that integrating some of the ChromBPNet data with eQTL datasets can be helpful. There are certainly limitations to this as our data is trained at a cell type and vascular site-specific resolution, and this level of resolution is lost in eQTL datasets. However, to address this suggestion, we aimed to prioritize the variant we had called out as an example, rs11677832 at the *COL6A3* locus. Our chromBPNet data predicts the alternative allele A to be associated with a closed chromatin state, this would suggest the variant allele would lead to decreased expression of *COL6A3*, an effect we predict is highest in SMCs and greater in descending SMCs over ascending SMCs. By interrogating the GTEx consortium data, we identified that rs11677832 is indeed an eQTL for *COL6A3* and the alternative allele A is associated with decreased expression in human aortic tissue. This perfectly fits with our data. We have now updated text to reflect this and have added an additional panel to figure 8. The updated text below also reflects significant modifications based on the following reviewer comment that is included below.

Page 27, line 609

“Following linear regression analysis of the abs_logFC within cell type between ascending and descending models (i.e. Asc SMC vs Desc SMC), we identified the top 1% of deviant SNPs (furthest away from linear regression line) based on quantile residuals. These top 1% deviant SNPs (total 274 SNP) highlight specific variants that appear to have vascular site-specific effect, and the 1% cut-off corresponds to a studentized p-value residual value of 1.73E-4. For example, rs11677932 is the lead SNP (chr2:237315312:G:A) at the *COL6A3* locus, a gene we have previously highlighted as having differential expression within cell types across vascular sites. The variant effect is predicted to lead to epigenetic silencing; however, this effect is greatest in SMC, compared to fibroblast and endothelial cells, and within SMCs, this effect is greatest in descending SMC compared to ascending SMC (studentized outlier p-value = 1.94E-33)(**Fig 8I**). To then evaluate if our predicted silencing effect of this variant is reflected in larger human variant to gene expression databases, we evaluated the Genotype-Tissue expression (GTEx) database¹ to determine if this variant (rs11677832) was an eQTL for *COL6A3*. We observed that rs11677832 is an eQTL for *COL6A3* (pval 6.75e-8) where the alternative allele A was associated with decreased *COL6A3* expression (**Fig 8J**), consistent with our predictive modeling suggesting epigenetic silencing. To then defined the proportion of loci predicted to have vascular site specific effect, we identified 84 of 115 loci (73%) from Pirruccello et al. (2022) study², to have either lead SNPs or SNPs in LD with the lead variant to be predicted to have vascular site specific effects in SMCs, 14 (12%) of which have 5 or more SNPs at the locus that meet our criteria for vascular site-specific effect. Top genes mapped to loci with vascular site-specific effect include *PRDM6*, *ULK4*, *ESR1*, *COL6A3*, *MSRA*, and *HAND2*.”

Updated Figure 8.

• What was the rationale or specific criterion used to define the top 1% of deviant SNPs selected for further analysis? On average, how many variants per GWAS locus exhibit vascular bed-specific differences in ChromBPNet scores, and what proportion of these also show vascular site-specific differences in chromatin accessibility? While the authors provide illustrative examples, it would be important to quantify the overall extent of

overlap across loci and report these summary statistics to assess the generalizability of the findings.

We chose to use the top 1% of deviant SNPs based on quantile residual score as this allowed us to specify top variants that have vascular site-specific effects. This corresponds to 274 SNPs. We have now computed the studentized two-way p-value to evaluate significance of deviation from linear regression modeling, and we have now defined that this 1% cutoff corresponds to a p value of $1.73E-4$. We then evaluated these 274 SNPs and identified that this represents 84 loci of the total 115 loci identified by Pirruccello et al to influence either ascending or descending thoracic aortic dimension.

To address this question, we have included the following paragraph as also noted above:

Page 27, line 609

“Following linear regression analysis of the *abs_logFC* within cell type between ascending and descending models (i.e. Asc SMC vs Desc SMC), we identified the top 1% of deviant SNPs (furthest away from linear regression line) based on quantile residuals. These top 1% deviant SNPs (total 274 SNP) highlight specific variants that appear to have vascular site-specific effect, and the 1% cut-off corresponds to a studentized p-value residual value of $1.73E-4$. For example, rs11677932 is the lead SNP (chr2:237315312:G:A) at the *COL6A3* locus, a gene we have previously highlighted as having differential expression within cell type across vascular site. The variant effect is predicted to lead to epigenetic silencing; however, this effect is greatest in SMC, compared to fibroblast and endothelial cells, and within SMCs, this effect is greatest in descending SMC compared to ascending SMC (studentized outlier p-value = $1.94E-33$)(**Fig 8I**). To then evaluate if our predicted silencing effect of this variant is reflected in larger human variant to gene expression databases, we evaluated the Genotype-Tissue expression (GTEx) database¹ to determine if this variant (rs11677832) was an eQTL for *COL6A3*. We observed that rs11677832 is an eQTL for *COL6A3* (pval $6.75e-8$) where the alternative allele A was associated with decreased *COL6A3* expression (**Fig 8J**), consistent with our predictive modeling suggesting epigenetic silencing. To then define the proportion of loci predicted to have

vascular site specific effect, we identified 84 of 115 loci (73%) identified from Pirruccello et al. (2022) study², to have either lead SNPs or SNPs in LD with the lead variant to be predicted to have vascular site specific effect in SMCs. 14 of which (12%) have 5 or more SNPs at the locus that meet our criteria for vascular site-specific effect. Top genes mapped to loci with vascular site-specific effect include *PRDM6*, *ULK4*, *ESR1*, *COL6A3*, *MSRA*, and *HAND2*.”

• **The attempt to link GWAS loci to DEGs is appreciated, but the chosen method, calculating the fraction of GWAS genes among DEGs, is not convincing. A gene being differentially expressed does not necessarily indicate that it mediates a GWAS signal; it may instead be co-expressed or a downstream consequence. A more appropriate approach would be to use established statistical frameworks such as LDSC or MAGMA, which test for enrichment of GWAS signals in gene sets.**

We agree that understanding cell types that mediate GWAS signals from single cell data is somewhat complex and established statistical frameworks such as MAGMA are useful tools. We have a particular interest in scDRS (Zhang et al., Nat Gen 2022: PMID 36050550) that can evaluate gene set enrichment analysis at single cell resolution and builds on the MAGMA method. We have now performed this analysis and included a paragraph in the manuscript in addition to two additional figure panels to figure 7.

Page 24, line 543

“To then understand how GWAS associated gene sets may have enriched expression within cell types, we performed polygenic enrichment analysis using single-cell disease relevance score (scDRS)³. First, we generated the individual raw gene expression matrix from scRNAseq data including groups determined by cell type and vascular site. With summary statistics GWAS data from Pirruccello et al. (2022)² we utilized the scDRS pipeline to evaluate GWAS gene enrichment for the ascending aortic dimension and descending aortic dimension GWASs. After lifting the human gene sets to the murine genome we then performed the scDRS test of group level statistics. We identified the associated Monte-Carlo P value (-log₁₀) to represent the significance of cell type-disease association for ascending aortic dimension and descending aortic dimension (Fig 7J-K). Here,

we observed that for the ascending aortic dimension GWAS (**Fig 7J**), the ascending SMC population had the strongest association, followed by carotid and descending aortic SMCs. Endothelial and fibroblast populations showed less strong of a relationship and macrophage cells had little to no association. The descending aortic dimension GWAS (**Fig 7K**) similarly showed ascending SMC as the primary cell type of association, however overall P value associations were lower, likely influenced by a smaller number of genomic loci associated with descending aortic dimension. These data implicate the vascular SMC to be the primary mediator of complex genetic influence on the “aortic dimension trait.”

Updated Figure 7

- Lines 418–420: “Other distinct fibroblast ascending GRN TF modules include Tcf21, Irf7/8, Sox4/7/17, and E2f8.” Please provide the figures in the Supplement.

We have now included a separate visualization to highlight some of these distinct Ascending Fibroblast GRNs now as supplemental figure 4, to provide more clarity for the reader.

New Figure S4:

Distinct Ascending Fibroblast GRN TFs:

E2f8, Egr2, Hes1, Hmgb2, Irf7, Irf8, Lef1, Mafb, Mecom, Meox1, Mkx, Myt1l, Npas3, Rbpj, Sox17, Sox4, Sox7, Tcf21, Tox, Trps1

Distinct Descending Fibroblast GRN TFs:
Nfatc2, Plagl1, Rorb, Sall1

- Lines 429–431, Figure 5H: The statement that “Meox1 has increased RNA expression in ascending fibroblasts, but this expression is largely limited to a specific cell population that represents valvular fibroblasts” seems incomplete. There is another population in the middle with high RNA expression—please clarify which cell type this corresponds to.

We agree that Meox1 expression by RNA is higher in ascending fibroblasts beyond just the valvular fibroblast population. The other cell population is also ascending aortic

fibroblasts that also highly express the adventitial fibroblast marker Pi16. This additional cell population reflects additional adventitial fibroblast cell clusters. We have now modified the text to include reference to the valvular fibroblast as well as other adventitial fibroblast cell populations.

- **Figure 5F: Define what the size and color of each dot represent in the figure legend.**

Done

- **Figure 7: Clarify the control group used. Ideally this is a random set of genes expressed in that cell type. This should be clearly stated in the Methods section.**

The control group reflects a random list of 100 genes that are expressed in the dataset, text has been modified to make this more clear for the reader.

- **Figure 8J–M: Please provide p-values for the enrichment results and indicate the number of top 1% deviant SNPs to allow assessment of robustness.**

This has now been included, please see response to the third comment above.

Reviewer #2 (Remarks to the Author):

The manuscript by Weldy et al., has been significantly revised relative to the first submission.

The “GWAS score” has been removed and a new computational analysis on the effect of candidate variants in specific cell-types and vascular sites added. I think the revision has added value to this work, which provides a high quality, rigorously generate dataset that can serve as “solid ground” for other investigators (as opposed for example to poor quality work Yu et al. mentioned in the rebuttal)

Even though the paper remains somewhat descriptive, it is a very valuable description, in my view.

In regard to the specific comments from editors and other reviewers:

[comment from revision 1]

1) Please provide the functional roles of some of the identified genes and clarify the clinical or biological relevance (reviewer #1, #2, #3)

[new comment from revision 2]

I think the authors have added valuable information with their new analysis on the predicted effect of previously identified variants affecting aortic size. The authors may consider commenting on variants of aortic dissection based on this pre-print doi: <https://doi.org/10.1101/2024.09.01.24312895>
Are genes involved in dissection expressed in different sites/cell-types?

We appreciate the reviewer in pointing to this manuscript, that we see has just been published a few weeks ago in JACC Advances (<https://www.jacc.org/doi/epdf/10.1016/j.jacadv.2025.101743>). This publication built multiple PRSs to predict thoracic aortic dissection, which used the Pirruccello data we had prioritized for one model, while also building additional models based on additional UKBB and MVP thoracic aortic dissection data and a 'GWAS by subtraction' method. The combined GWAS identifies 41 new loci associated with dissection. In looking at these loci, it appears some of these loci are also differentially expressed between ascending and descending thoracic aortic fibroblasts, one example is *ARHGAP31*. The genes involved in dissection also include the genes that Pirruccello had identified for aortic dimension, as they had also reported that their aortic dimension PRS predicted thoracic aortic dissection. This new data likely reflects further loci that influence disease risk in a cell type and vascular site-specific method. We have now added text to the discussion to cite this paper.

[comment from revision 1]

2) Please explain the differential TGF β response of cells in the ascending vs descending aorta and its physiological or pathological relevance (reviewer #1, #2)

[new comment from revision 2]

The work conducted in adventitial fibroblasts is certainly valuable, but it would be informative to include mention of a condition (or more) where differential response/TF activation in response to TGF-beta in adventitial fibroblasts of different vascular sites may be critical to pathogenesis, if known. Although a similar experiment with smooth muscle cells and endothelial cell would bring critical insight, I am aware that these cells are basically impossible to grow from mouse aorta.

We agree that the differential response to TGF β in adventitial fibroblasts is important, and in the manuscript, we highlight that the genes that drive hereditary thoracic aortic dissection disorders show distinct differential expression. We have now modified the text to specifically highlight Loeys-Dietz syndrome, that results in abnormal TGF β signaling and preferentially leads to thoracic aortic dissection.

Page 14 line 309 now reads:

“This is notably relevant to Loeys-Dietz syndrome, characterized by abnormal TGF β signaling that preferentially results in thoracic aortic disease.”

[comment from revision 1]

3) Please provide stronger support to a major claim that different vascular beds retain an “epigenetic memory” depending on the origin of the vascular cells, by comparing the gene accessibility of the mature vascular cells of each vascular bed with those of their developmental lineage (as per R#3’s advice), taking into account (or ruling out) the role of different hemodynamics in shaping up the epigenome of each vascular bed.

[new comment from revision 2]

**The manuscript has been reworded to account for this issue—
In any case, from a biomedical point of view, what truly matters is that these cells have different epigenetic memory/chromatin accessibility profiles depending on where they are from in the arterial tree—because this may affect pathophysiology of specific cardiovascular conditions. Even lineage tracing by progenitors would not necessarily rule out “environmental exposure” because location (i.e. closer to lumen vs adventitia for CNC/SHF) would still be a variable.**

We agree that understanding developmental lineage tracing versus anatomic location are related questions however remain somewhat distinct. Our hope from our manuscript was to understand how anatomic location influences epigenetic biology within the vascular wall and this is an ongoing area of research.

[comment from revision 1]

**4) Please address all technical issues raised, including (but not limited to):
a) increasing the number of samples used for the scRNA-Seq and ATAC-Seq
b) providing more details regarding the data analysis and additional data on quality control, the number of unique molecular identifiers (UMIs), transcripts, and peaks identified in the supplement (Reviewers #1, #2, #3)
c) engaging more advanced methods to integrate both scATAC and scRNA-Seq data and present the validation of some of the predicted TFs (reviewer #1).**

[new comment from revision 2]

I think the authors have addressed this points.

Minor

Typo on line 637 “vasculitides”

‘Vasculitides’ is actually the plural of ‘vasculitis’. For example, a sentence may read, “A physician has distinct expertise across a wide range of vasculitides.”

Reviewer #3 (Remarks to the Author):

Weldy et al resubmission of “The epigenomic landscape of single vascular cells reflects developmental origin and identifies disease risk loci” includes a larger dataset of single cell RNA and ATAC-seq. Now more mice are included, resulting in an increase in an increase in cells for scRNA-seq from 21k to 37k and for scATAC-seq from 23k to 40k. This does not substantially change the results for the description of cellular heterogeneity in the beginning of the paper. The prior submission used a GWAS score to show disease relevance for differentially expressed genes. This submission removes that section entirely. Instead there is a novel ChromBPNet algorithm that uses a machine learning approach to prioritize variants that affect TF binding. This data is presented at a high level, and though interesting, does not provide much validation of disease relevance for the differential gene analysis in the paper.

1. The authors state in their rebuttal that the prior Pirruccello et al, Nature Genetics 2022 paper, “was based on cMRI data and GWAS data from the UKBB and did not include any gene expression analysis.” Figure 4 of that paper is scRNA-seq analysis from 4 human ascending and descending aortas. They conduct cell-type specific gene expression analysis for all causal genes prioritized by TWAS (using GTEx data) in the ascending and descending aorta. Since this analysis is from human tissue instead of mice, I think the genes prioritized by this earlier approach are more valid for functional follow-up experiments.

We sincerely apologize as we mistakenly forgotten about the Pirruccello work on TWAS signal and single cell data. The reviewer is correct that figure 4 of that paper is scRNAseq analysis and they performed a TWAS analysis. Our hope was to map the broader epigenomic landscape, and our manuscript is the first to do so across vascular sites and related to human genetics data from our knowledge.

2. The ChromBPnet data seems to detract from the conclusions of the paper. The 7 scatter plots seem to show strong correlation between cell-types from different sites or lineages. The authors suggest that the SMC scatter plot shows greater variation, but no statistical analysis is used to bolster this claim. In fact, all 7 plots seem to show high correlation. I do not think this new analysis addresses the concerns raised with the GWAS score data in the first submission.

We appreciate the reviewer's comment and have now included additional P-value data and better discussion of the statistical analysis used. Please see comment #3 from reviewer 1.

3. Other suggestions for proving the epigenetic memory hypothesis such as comparison with developmental cell types were not conducted.

Type line 560

Figure 4: figures K and L are out of order

Corrected.

References

1. Consortium GT. The Genotype-Tissue Expression (GTEx) project. *Nat Genet.* 2013;45:580-585. doi: 10.1038/ng.2653
2. Pirruccello JP, Chaffin MD, Chou EL, Fleming SJ, Lin H, Nekoui M, Khurshid S, Friedman SF, Bick AG, Arduini A, et al. Deep learning enables genetic analysis of the human thoracic aorta. *Nat Genet.* 2022;54:40-51. doi: 10.1038/s41588-021-00962-4
3. Zhang MJ, Hou K, Dey KK, Sakaue S, Jagadeesh KA, Weinand K, Taychameekiatchai A, Rao P, Pisco AO, Zou J, et al. Polygenic enrichment distinguishes disease associations of individual cells in single-cell RNA-seq data. *Nat Genet.* 2022;54:1572-1580. doi: 10.1038/s41588-022-01167-z

13th Aug 2025

Manuscript Number: MSB-2025-13237-T

Title: Epigenomic landscape of single vascular cells reflects developmental origin and disease risk loci

Author: Chad Weldy

Soumya Kundu

João Monteiro

Wenduo Gu

Albert Pedroza

Alex Dalal

Matthew Worssam

Daniel Li

Brian Palmisano

Quanyi Zhao

Disha Sharma

Trieu Nguyen

Ramen Kundu

Michael Fischbein

Jesse Engreitz

Anshul Kundaje

Paul Cheng

Thomas Quertermous

Dear Chad,

Thank you for submitting your revised manuscript to Molecular Systems Biology. We have now received the enclosed report from the reviewer who reviewed the previous version of your manuscript at the other journal. As you will see below, the Reviewer is satisfied with the revisions made and thinks the study is suitable for publication.

Therefore, I am pleased to inform you that we are able to accept your manuscript, pending the following revision:

1. Please submit the manuscript in .docx format.
2. Please upload each figure as an individual high-resolution file. Remove all figures from the manuscript file, but keep the figure legends in place below the References section.
3. Figure callouts should appear in sequential order. Please add the missing callout for Figure 8N.
4. Please provide up to five keywords in the manuscript file.
5. Please remove the "Author contributions" section from the manuscript.
6. Data availability: Please provide specific URLs for GSE296197 and GSE296074 datasets in the Data Availability statement.
7. "Disclosure" should be renamed to "Disclosure and Competing Interests Statement".
8. Remove the section heading "Funding", and the related funding information need to be moved to the Acknowledgment section.
9. The references need to be formatted according to the Molecular Systems Biology reference style. Please list up to 10 co-authors of a paper before adding et al. in the reference list. Citations should be listed in alphabetical order.
10. Appendix:
 - The Appendix file needs to be submitted in PDF format.
 - The title page should read "Appendix for [manuscript title]" instead of "Supplementary Figures" and it should contain a Table of Content with the page numbers for all listed items.
 - nomenclature should be "Appendix Figure Sx" consistently throughout both the manuscript and the Appendix PDF
11. Supplementary tables: Update the source file names, titles, legends, and manuscript callouts to use "Dataset EV1-EV25" instead of "Supplemental Table 1-25." Legends should be included in a separate tab (sheet) within each corresponding Excel file.

12. All Materials and Methods need to be described in the main text using our 'Structured Methods' format. According to this format, the Methods section includes a Reagents and Tools Table (listing key reagents, experimental models, software and relevant equipment and including their sources and relevant identifiers) followed by a Methods and Protocols section describing the methods, ideally using a step-by-step protocol format. The aim is to facilitate adoption of the methodologies across labs. Please download and fill our Reagents and Tools Table template (.docx), which you can find in our author guidelines: <https://www.embopress.org/page/journal/17444292/authorguide#structuredmethods>.

13. Please provide a "standfirst text" summarizing the study in one or two sentences (approximately 250 characters, including space), three to four "bullet points" highlighting the main findings and a "visual abstract" (550px width and 400-600 px height, PNG format) to highlight the paper on our homepage.

Here are a couple of examples:

<https://www.embopress.org/doi/10.15252/msb.20199356>

<https://www.embopress.org/doi/10.15252/msb.20209475>

<https://www.embopress.org/doi/10.15252/msb.209495>

15. When you resubmit your manuscript, please download our CHECKLIST (<https://www.embopress.org/pb-assets/embosite/EMBO%20Press%20Author%20Checklist-1642513524327.xlsx>) and include the completed form in your submission.

Please note that the Author Checklist will be published alongside the paper as part of the transparent process (<https://www.embopress.org/page/journal/17444292/authorguide#transparentprocess>).

16. Please address the following issues related to figure legends:

- Please indicate the statistical test used for data analysis in the legends of figures 3A, 4D, 6C-F; 7A, B, D, E, G, H

- Please note that the box plots need to be defined in terms of minima, maxima, centre, bounds of box and whiskers, and percentile in the legend of figure 8J

- Please note that information related to n is missing in the legends of figures 2J, R; 3A, 4K, L, N, P; 6C-F; 7A, B, D, E, G, H

17. Section headings should be renamed and reordered as follows: Title page - Abstract - Keywords - Introduction - Results - Discussion - Methods - Data Availability - Acknowledgements - Disclosure and Competing Interests Statement - References - Figure Legends - Table(s) - Expanded View Figure Legends.

Click on the link below to submit your revised paper.

Kind regards,
Jingyi

Jingyi Hou, PhD
Senior Editor
Molecular Systems Biology

If you do choose to resubmit, please click on the link below to submit the revision online before 12th Sep 2025.

As a matter of course, please make sure that you have correctly followed the instructions for authors as given on the submission

website.

*** PLEASE NOTE *** As part of the EMBO Press transparent editorial process initiative (see our Editorial at <https://dx.doi.org/10.1038/msb.2010.72> , Molecular Systems Biology will publish online a Review Process File to accompany accepted manuscripts. When preparing your letter of response, please be aware that in the event of acceptance, your cover letter/point-by-point document will be included as part of this File, which will be available to the scientific community. More information about this initiative is available in our Instructions to Authors. If you have any questions about this initiative, please contact the editorial office (msb@embo.org).

Reviewer #1:

I would like to thank the authors for their detailed response, which addresses my concerns. While experimental validation would be valuable, I accept the authors' rationale for not pursuing it in this manuscript. In my view, the manuscript is now suitable for publication.

All editorial and formatting issues were resolved by the authors.

19th Aug 2025

Manuscript number: MSB-2025-13237R

Title: Epigenomic landscape of single vascular cells reflects developmental origin and disease risk loci

Dear Dr Weldy,

Thank you again for sending us your revised manuscript. We are now satisfied with the modifications made and I am pleased to inform you that your paper has been accepted for publication.

Kind regards,
Jingyi

Jingyi Hou, PhD
Senior Editor
Molecular Systems Biology
